

# Petrogenesis of Early Paleozoic I-type granitoids in the Longshoushan and implications for the tectonic affinity and evolution of the southwestern Alxa Block

Renyu Zeng [1, 2, 3], Hui Su [1], Mark B. Allen [3], Haiyan Shi [4], Houfa Dua, [1, 2], Chenguang Zhange [5], Jie Yan [1, 2]

[1] School of Earth Sciences, East China University of Technology, Nanchang, 330013, China
[2] State Key Laboratory of Nuclear Resources and Environment, East China University of Technology, Nanchang 330013, China
[3] Department of Earth Sciences, Durham University, Durham DH1 3LE, UK
[4] Qinghai Geological Survey, Technology Innovation Center for Exploration and Exploitation of Strategic Mineral Resources in Plateau Desert Region, Ministry of Natural Resources, Xining 810000, China
[5] School of Geographic Sciences, Xinyang Normal University, Xinyang, 464000, China

*Correspondence to*: Renyu Zeng (zengrenyu@126.com)

**Abstract.** In the Paleozoic, the Alxa Block was situated between the Central Asian Orogenic Belt and the North Qilian Orogenic Belt, and experienced intense magmatic activity. Thus, the Alxa Block is an important area for understanding the tectonic framework and evolution of these two orogenic belts. However, there has long been debate regarding the tectonic affinity and tectonic evolution of the Longshoushan, located in southwestern margin of the Alxa Block, during the Paleozoic. In this study, we present zircon U-Pb ages, whole-rock major and trace elements, and Hf isotopic data for the granitoids from the east of the Longshoushan to investigate these issues. Bulk-rock analyses show that these granitoids are weakly peraluminous, with high $SiO_2$ and $K_2O$, but low MgO, $TFe_2O_3$ and $P_2O_5$. They are also characterized by enrichment in LREE and LILE, depletion in HREE and HFSE, and a large range of variation in $\varepsilon Hf(t)$ values (monzogranite: -0.37 to -16.28; K-feldspar granite: 3.53 to -7.74). These geochemical features indicate that these granitoids are highly fractionated I-type granite, which were formed by crust and mantle-derived magma mixing. LA-ICP-MS zircon U-Pb dating constrains that the monzogranite and K-feldspar granite were formed at 440.8 ±2.1 Ma and 439.4 ± 2.0 Ma, respectively. Combining these results with previous chronological data, the geochronology framework of Paleozoic magmatic events in the Longshoushan is consistent with the North Qilian Orogenic Belt to the south, but significantly differs from other parts of the Alxa Block and the Central Asian Orogenic Belt to the north. This result indicates that the Longshoushan was primarily influenced by the North Qilian Orogenic Belt during the Early Paleozoic. Integrated with previous studies, a three-stage tectonic model is proposed of Early Paleozoic accretion and arc magmatism leading to collision in the Longshoushan. (1) 460-445 Ma: Arc magmatism on an active continental margin with the northward subduction of the North Qilian back-arc basins (NQ bab). (2) 445-435 Ma: Magmatic rocks, dominated by I-type granites, were formed in a continent-continent collision setting. Significant crustal thickening is interpreted to result from compressional stress and/or magmatic additions. (3) 435-410 Ma: The





development of abundant A-type granites and mafic dikes in response to intraplate extension, supported by a change in trace
element chemistry indicating crustal thinning at this stage. This sequence of events and their timings is similar to other parts
of the Central China Orogenic Belt, and requires either a coincidence of several oceanic plates closing at the same time, or an
along-strike repetition of the same system.

## 1 Introduction

The Alxa Block is a region of Precambrian basement adjacent to Phanerozoic orogenic belts. Its northern and south sides are
adjacent to the Central Asian Orogenic Belt and the North Qilian Orogenic Belt, respectively (Song et al., 2013; Xue et al.,
2017; Zeng et al., 2016, 2021; Hui et al., 2021; Allen et al., 2023). The eastern margin may have undergone collision and
amalgamation events with the North China Craton, although the timing of the amalgamation is unclear (Wang et al., 2015;
Dan et al., 2016). The Alxa Block therefore documents the tectonic evolution of the Proto-Tethys and Paleo-Asian oceans.

The Longshoushan is a mountain range located in the south part of the Alxa Block and is characterized by Early Paleozoic
magmatic activity. It is generally believed that the region to the north part of the Longshoushan, known as the Beidashan, is
mainly influenced by the Central Asian Orogenic Belt (Liu et al., 2016; Zhou et al., 2016), while the region to the south, the
Hexi Corridor, is influenced by the North Qilian Orogenic Belt (Wei et al., 2013; Zhang et al., 2017). However, the tectonic
background of the Longshoushan itself is still debated. Some researchers suggest that the Early Paleozoic magmatic rocks in
the Longshoushan are related to the subduction-accretionary orogenesis of the North Qilian Orogenic Belt (Liu et al., 2021;
Zeng et al., 2021), while others propose a connection between these magmatic events and the roll-back of the Paleo-Asian
Oceanic slab (Liu et al., 2016; Zhou et al., 2016; Xue et al., 2017). Furthermore, it is widely believed that the Longshoushan
underwent a stress transition from a compressional to an extensional environment during the Early Paleozoic (Zeng et al., 2016;
Zhang et al., 2017; Wang et al., 2018; Liu et al., 2021). However, there are different viewpoints regarding the age of this
transition, with some suggesting it occurred in the Late Ordovician (ca. 450-440 Ma, Zhang et al., 2017; Liu et al., 2021) and
others proposing it took place in the Early Silurian (ca. 433-430 Ma, Yu et al., 2015; Zeng et al., 2016). Therefore, there is still
considerable debate regarding the tectonic background and tectonic evolution of the Longshoushan during the Early Paleozoic.

As an important component of the continental crust, granitoid is of great significance in studying crustal properties,
tectonic framework and tectonic evolution (e.g. Pearce et al., 1984; Zeng et al., 2022). Extensive research has been conducted
on granitoids in the Longshoushan, including Zhimengou, Taohualashan, Jiling, Jinchuan, Qingshanbao, and the western
extension of the Helishan area. However, research on the granitoids in the east of the Longshoushan has not yet been
undertaken. In this contribution, whole-rock geochemistry, zircon U-Pb ages, and Hf isotopes of the granitoids in the east of
the Longshoushan are systematically studied. In conjunction with previously published data on Paleozoic magmatic rocks in
the Longshoushan, this study also aims to investigate the tectonic background, crustal thickness variations, and tectonic
evolution of the Longshoushan during the Early Paleozoic.



## 2 Geological setting

The Alxa Block is separated from the Hexi Corridor and the Qilian Orogenic Belt by the Longshoushan Fault to the south, from the Tarim Craton by the Altyn Fault to the west (Zhang and Gong, 2018), from the North China Craton by either the Bayanwulashan Fault, western marginal fault of the Ordos Basin or the Helanshan Fault to the east (Hui et al.,2021) (Fig. 1). On the northern margin of the Alxa Block, there are two important ophiolite belts: the Engger Us Ophiolite Belt in the north

and the Qagan Qulu Ophiolite Belt in the south. Hf isotope data show that the magma sources of the Paleozoic magmatic rocks on both sides of the Qagan Qulu Ophiolite Belt are different (Zhang et al., 2015). Hence, recent studies have indicated that the Qagan Qulu Ophiolite Belt was most likely a Late Paleozoic suture related to the closure of a back-arc basin, representing the tectonic boundary between the Alxa Block and the Central Asian Orogenic Belt (Zhang et al., 2015; Hui et al., 2021).

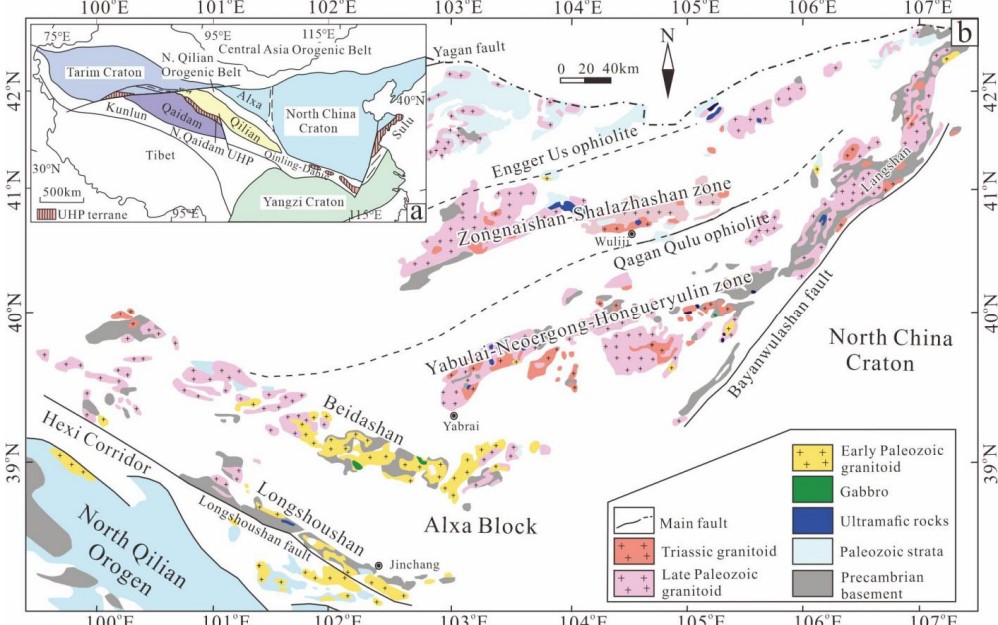

**Figure 1 (a) Sketch geological map showing the tectonic location of the Alxa Block (modified from Song et al., 2013). (b) Simplified geological map of the Alxa Block Phanerozoic granitoids (modified from Zheng et al., 2014; Hui et al., 2021). AB- Alxa Block; CQB- Central Qilian Block; NCC- North China Craton.**

The tectono-thermal events in the Alxa Block mainly occurred during the Paleoproterozoic and Paleozoic to early Mesozoic. During the Paleoproterozoic, the Alxa Block experienced the ~2.5 Ga magmatic-metamorphic event, ~2.3 Ga and

2.05-2.0 Ga magmatic events, as well as the 1.95-1.80 Ga magmatic-metamorphic event (Zhang et al., 2013; Gong et al., 2016; Zeng et al., 2018; Qi et al., 2019). The ~2.5 Ga magmatic-metamorphic event is relatively limited in distribution and is mainly reported in the Beidashan (Zhang et al., 2013) and the Longshoushan (Qi et al., 2019). The ~2.3 Ga and 2.05-2.0 Ga magmatic



events are primarily found in the Bayanwulashan, Diebusuge and Longshoushan areas, and they are generally believed to be related to an extensional tectonic setting (Zeng et al., 2018). The 1.95-1.80 Ga metamorphic events are widely documented in
the metamorphic basement throughout the Alxa Block (Zhang et al., 2013; Gong et al., 2016; Zeng et al., 2018). It is generally believed to be associated with a late Paleoproterozoic orogeny (Gong et al., 2016; Zeng et al., 2018), possibly recording the amalgamation of the North China Craton (Zhang et al., 2013; Zeng et al., 2018).

During the Paleozoic to Early Mesozoic, the Alxa Block was mainly influenced by the Central Asian Orogenic Belt to the north and the North Qilian Orogenic Belt to the south, resulting in extensive magmatic activities. Zhang and Gong (2018)
divided the magmatic activities during this period into three stages: 460-390 Ma, 360-300 Ma, and 299-230 Ma. These magmatic activities in the Alxa Block generally exhibited a progressive younging trend from south to north. Compared to the other two stages, the 460-390 Ma magmatic events were relatively limited in distribution within the Alxa Block. They were mainly exposed in the central-southern parts of the Alxa Block, such as the Beidashan and Longshoushan areas, as well as in the eastern regions like the Yamatu and Langshan areas. The tectonic settings of the 460-390 Ma magmatism in the Alxa Block
are not fully understood. Some researchers believe that 460-390 Ma magmatic events are related to the subduction of oceanic crust related to the Central Asian Orogenic Belt (Liu et al., 2016; Zhou et al., 2016; Xue et al., 2017). For example, Xue et al. (2017) suggest that these rocks are a result of the slab rollback of the Paleo-Asian Oceanic slab, while Zhou et al. (2016) propose a connection with post-arc rifting. However, some work suggests that the 460-390 Ma magmatic events in the southern margin of the Alxa Block, particularly in the Longshoushan, are related to the North Qilian Orogenic Belt (Zeng et al., 2016,
2021; Zhang et al., 2017; Liu et al., 2021). Additionally, some researchers suggest that the 460-390 Ma magmatic events in the eastern part of the Alxa Block are the product of collision and accretion between the Alxa Block and the North China Craton (Wang et al., 2015; Dan et al., 2016).

The magmatic events during the periods of 360-300 Ma and 299-230 Ma are widely distributed in the Alxa Block, especially in the Yabulai-Nuorigong-Honggueryulin zone and the Beidashan. The predominant rock type is granite, with a
small amount of gabbro also present (e.g., Zhang and Gong, 2018; Dan et al., 2014; Zhang et al., 2016). These two periods of magmatic events are generally believed to be associated with the subduction-collision and post-collisional evolution of the Central Asian Orogenic Belt (Zhang and Gong, 2018; Liu et al., 2017).

## 3 Samples and petrography

The investigated plutons in this study are located in the east of the Longshoushan (Fig. 1b), including monzogranite and K-
feldspar granite. In the study area, the early Precambrian Longshoushan Complex, Neoproterozoic Dunzigou Formation, Hanmushan Formation and Neogene strata are well exposed (Fig. 2). The Longshoushan Complex is mainly composed of biotite quartz schist, biotite quartz schist and migmatite, while the Dunzigou Formation and Hanmushan Formation are marine clastic and carbonate sedimentary rocks. The monzogranite and K-feldspar granite both intruded into the Early Precambrian



Longshoushan Complex (Fig. 2). Four samples of monzogranite and four samples of K-feldspar granite were collected at

locations shown in Fig. 2b.

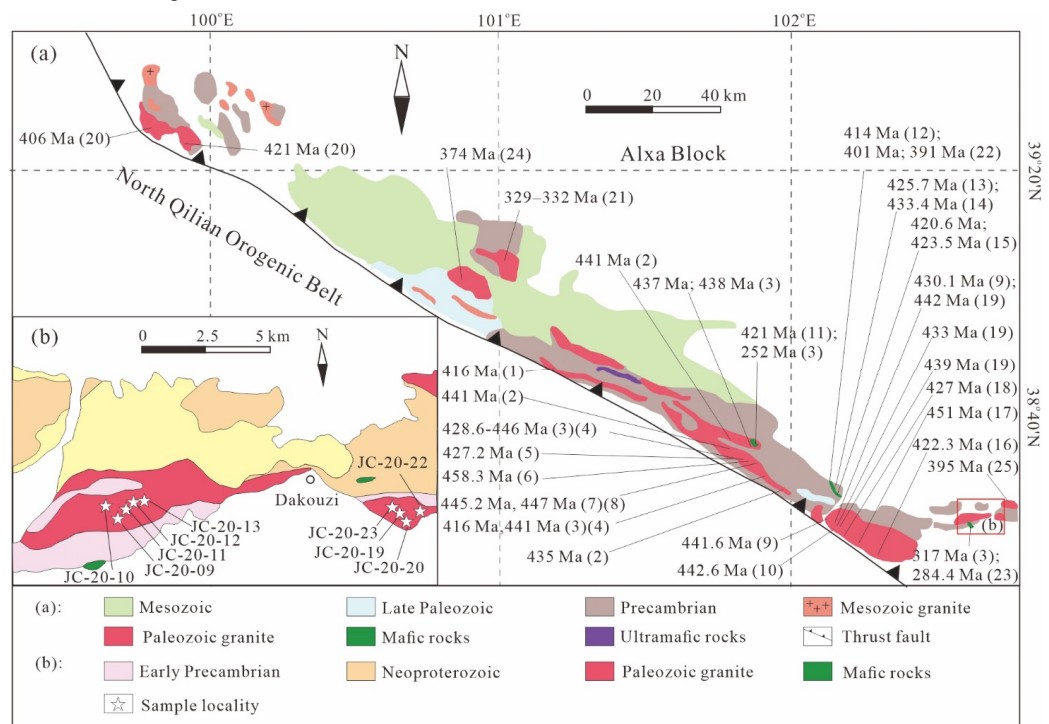

**Figure 2 (a) Simplified geological map of the southwestern Alxa Block (Wang et al., 2020); (b) Simplified geological map of the east of the Longshoushan area. Data are in Supplementary materials Table S1.**

### 3.1 Monzogranite

The monzogranite is generally pale red in colour with fine to medium grained texture (Fig. 3a). The mineral assemblage contains K-feldspar (~32 %), plagioclase (~28 %), quartz (~35 %) and biotite (~5 %) as well as accessory minerals such as magnetite, zircon and apatite. K-feldspar grains are euhedral or subhedral, always exhibit grid twinning (Fig. 3b) and can contain rounded quartz and plagioclase inclusions (Fig. 3c). Plagioclase commonly show polysynthetic twinning, and are altered to sericite and carbonate, with rims surrounded by newly formed K-feldspar (Fig. 3b). Biotite is the main dark mineral,

occurring in flaky shapes and filling in the interstices between the other minerals, with embayment features and chloritization (Fig. 3b).



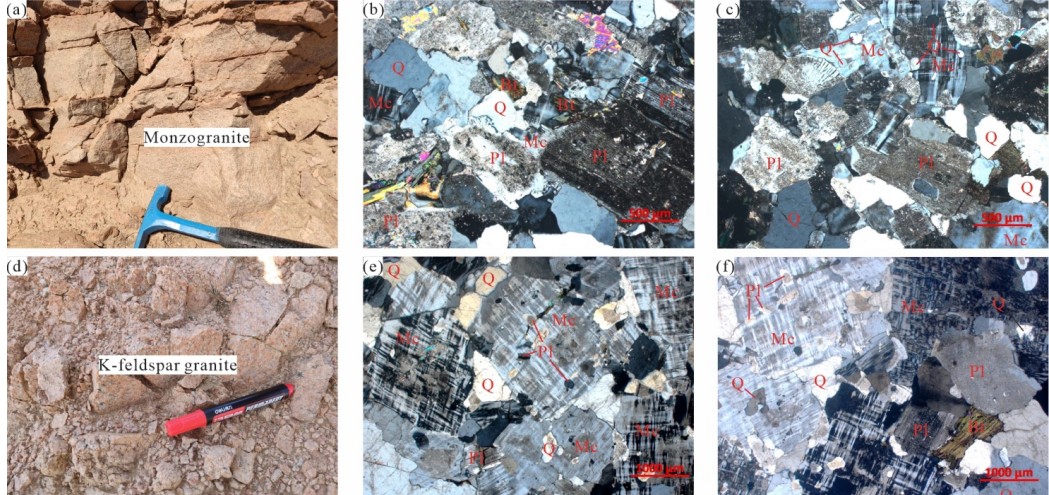

**Figure 3 Field photographs and photomicrographs of monzogranite (a, b, c) and K-feldspar granite (d, e, f) (b, c, e, f-perpendicular polarized light). Q-quartz; Kfs-feldspar; Pl- plagioclase; Mc-Microcline; Bt-biotite.**

### 3.2 K-feldspar granite


The K-feldspar granite is generally pale red in colour with medium grained texture (Fig. 3d). The mineral assemblage contains K-feldspar (~55 %), quartz (~30 %), plagioclase (~10 %) and biotite (~5 %) as well as accessory minerals such as apatite, magnetite and zircon. K-feldspar has a grain size range between 3mm to 4 mm, mainly composed of microcline with grid twinning, and containing numerous plagioclase and quartz inclusions (Fig. 3e). Plagioclase usually exhibits polysynthetic

twinning and has sericitization and carbonatization. Euhedral to subhedral crystals are tabular-shaped. The main dark mineral is biotite; grains have embayed crystal margins and chloritization (Fig. 3f).

### 4 Analytical methods

Representative rock samples were selected for geochemical analysis. The analyses of whole-rock major and trace elemental compositions were conducted at Analytical Chemistry & Testing Services (ALS) Chemex (Guangzhou) Ltd, by using wave-

dispersive X-ray fluorescence (XRF) (ME-XRF26) and ICP-MS (ME-MS81), respectively. The detailed methods were given by Zeng et al. (2022).

Cathodoluminescence (CL) imaging of zircon were performed at Chengpu Geological Testing Co. Ltd, Langfang, China using the TIMA analysis. The LA-ICP-MS zircon U-Pb dating and trace element analyses were carried out at Wuhan Sample Solution Analytical Technology Co., Ltd., Wuhan, China. Laser ablation system is the COMPexPro 102 ArF excimer laser

(wavelength of 193 nm and maximum energy of 200 mJ) with a spot size of 35 μm, repetition rate at 8 Hz. ICP-MS is Agilient



7900.Zircon 91500 and glass NIST610 were used as external standards for U-Pb dating and trace element calibration, respectively. CJ-1 and Plešovice were used for quality control. In the experiment, the weighted mean age of GJ-1 and Plešovice were $600.3 \pm 3.7$ Ma (n=7, MSWD = 0.04) and $337.5 \pm 3.0$ Ma (n=4, MSWD = 0.03), respectively. The detailed methods were given by Zong et al. (2017).

Zircon in-situ Lu–Hf analyses were undertaken at the Wuhan Sample Solution Analytical Technology Co., Ltd, Hubei, China using a Neptune Plus MC-ICP-MS. An ArF excimer laser ablation system of GeoLas HD was used with 44 μm spot size. The detailed analytical program is the same as outlined by Hu et al. (2012). During our analyses, Plešovice, 91500 and GJ-1 as standard zircons has the values of 0.282472–0.282495, 0.282302–0.282314 and 0.282024–0.282032 respectively, consistent with their recommended values (Plešovice: $0.282482 \pm 0.000023$; 91500: $0.282308 \pm 0.0000106$; GJ-1: $0.282010 \pm$

0.0000089, Zhang et al., 2020).

## 5 Analytical results

The data for major and trace elements, zircon U-Pb ages, zircon trace elements, and zircon Hf isotopes are shown in Tables S2, S3, S4 and S5, respectively.

### 5.1 Geochemical characteristics

#### 5.1.1 Monzogranite

The samples of monzogranite have $SiO_2$ contents ranging from 71.59 wt.% to 72.05 wt.%. They have low $Na_2O$ (3.75–3.88 wt.%) and high $K_2O$ (4.40–4.52 wt.%). All samples fall in the granite area in the TAS classification (Fig.4a), and the Calc-alkaline series area in the $SiO_2$-$N_2O$+$K_2O$ diagram (Fig.4b). Contents of $Al_2O_3$ are 14.40–14.47 wt.%, A/KNC are 1.05–1.07. All compositions fall in the weakly peraluminous field of the A/NC-A/KNC diagram (Fig.4c).



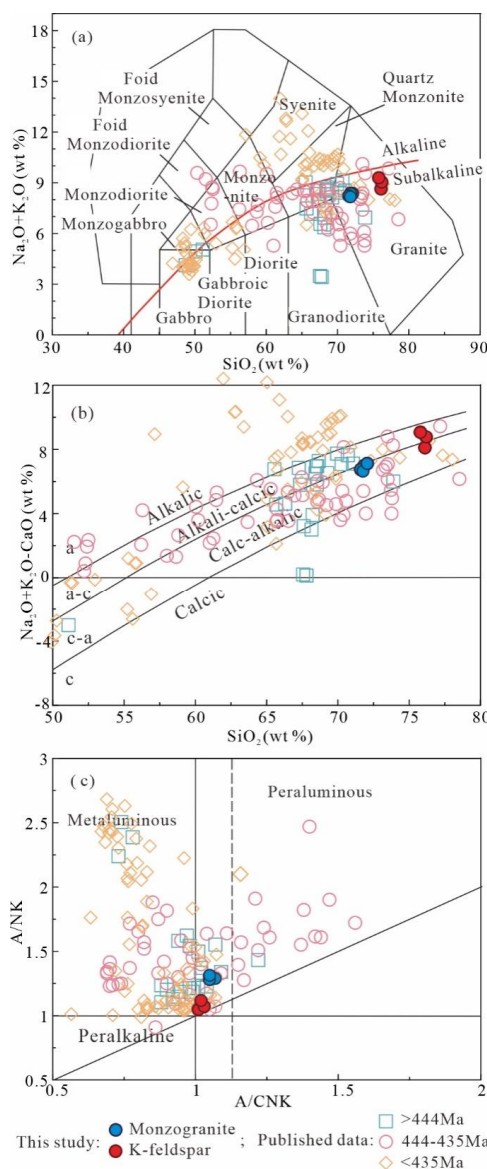


**Figure 4 Geochemical classification diagrams. (a) TAS diagram (after Middlemost, 1994); (b) A/CNK-A/NK diagram (after Maniar and Piccoli, 1989); (c) Na₂O+K₂O-CaO vs. SiO₂ diagram (after Frost et al., 2001). The source of the published data can be found in Supplementary materials Table S1.**



The La$_N$/Yb$_N$ values of monzogranite range from 22.26 to 27.84, showing enrichment of LREE and depletion of HREE.
On chondrite–normalized REE patterns (Fig. 5c), the monzogranite has relatively inclined LREE patterns and flat HREE patterns. All the samples display negative Eu anomalies (δEu=0.73–0.75) and slight positive Ce anomalies (δCe=0.99–1.22). In the primitive mantle-normalized trace element diagram (Fig. 5d), these samples show enrichment of LILEs (e.g., Rb and K), depletion of HFSEs (e.g., Nb, Ta, Ti and P), and no depletion of Hf and Zr.

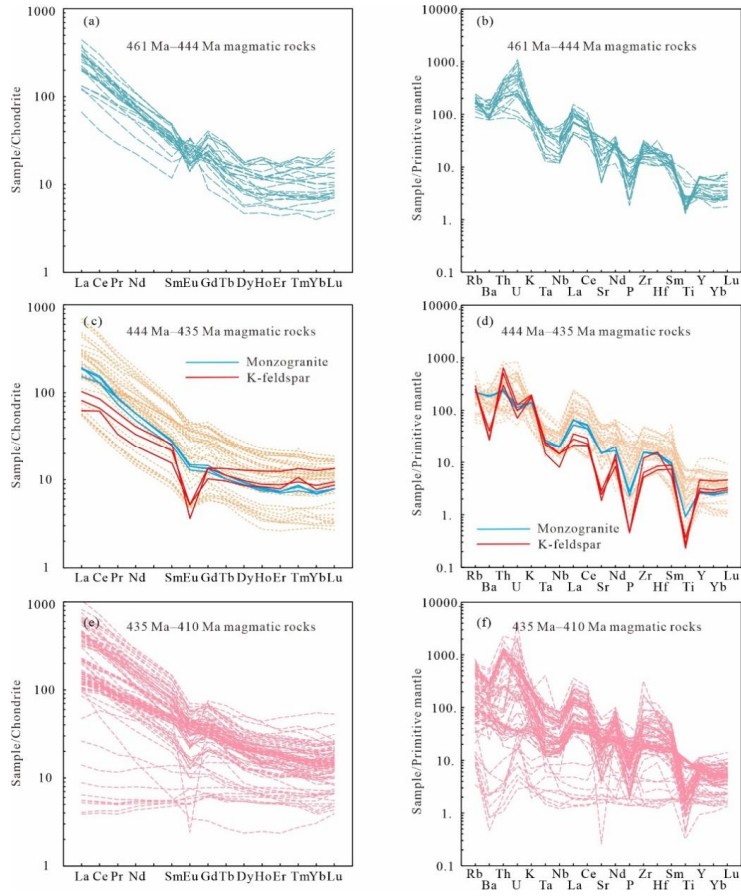

**Figure 5 Chondrite-normalized REE patterns and primitive mantle-normalized trace element patterns of the Dakouzi granite (chondrite and primitive mantle values are from Sun and McDonough, 1989). The source of the published data can be found in Supplementary materials Table S1.**




### 5.1.2 K-feldspar granite

The samples of K-feldspar granite have $SiO_2$ contents from 75.77 wt.% to 76.18 wt.%. They have low $Na_2O$ (3.09–3.13 wt.%)
and high $K_2O$ (5.54–6.18 wt.%). All samples fall in the granite area in the TAS classification (Fig.4a), and the Alkali -calcic
series and Calc-alkaline series areas in the $SiO_2$-$N_2O$+$K_2O$ diagram (Fig.4b). Contents of $Al_2O_3$ are 12.33–12.36 wt.%, A/KNC
are 1.01–1.03, and all samples are weakly peraluminous in the A/NC-A/KNC diagram (Fig.4c).

The $La_N/Yb_N$ values of K-feldspar granite samples are from 6.38 to 11.94, showing enrichment of LREE and depletion
of HREE. On chondrite–normalized REE patterns (Fig. 5c), the K-feldspar granite has relatively inclined LREE patterns and
flat HREE patterns. All samples display clear negative Eu anomalies ($\delta Eu$=0.19–0.41) and slight positive Ce anomalies
($\delta Ce$=1.08–1.35). In the primitive mantle-normalized trace element diagram (Fig. 5d), these samples show enrichment of
LILEs (e.g., Rb, Ba and K), strong depletion of HFSEs (e.g., Nb, Ta, Ti and P), and no depletion of Hf and Zr.

### 5.2 Zircon U-Pb geochronology, trace elements and Hf isotopes

### 5.2.1 Monzogranite

Zircons from monzogranite (JC-20-9) are mostly euhedral, transparent, colorless, and 80–160 μm in length. The zircons have
bright CL intensity and oscillatory zoning in the CL images (Fig. 6a), with Th/U ratios of 0.10–1.27. 25 spots cluster on the
concordia curve with $^{206}Pb/^{238}U$ ages of 429–449 Ma, defining a weighted mean age of 440.8±2.1 Ma (MSWD=1.6) (Fig. 6a,
b).

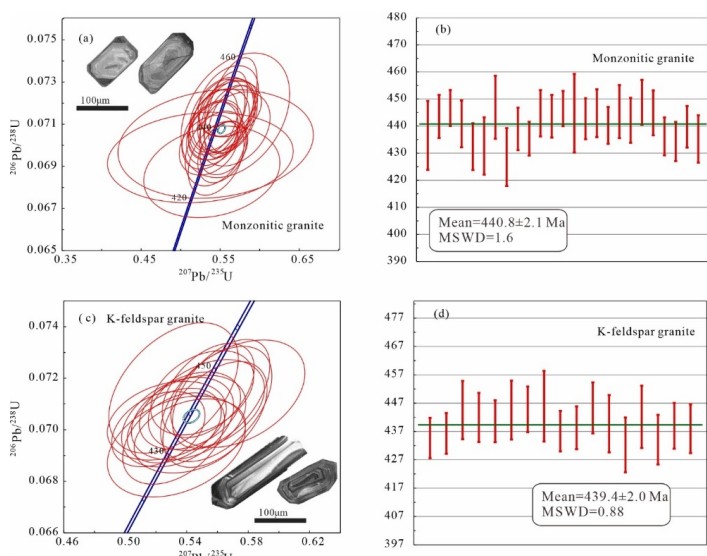

**Figure 6 Concordia diagrams for zircon LA-ICP-MS U–Pb analyses.**



Sixteen zircons were analyzed for REE. The spots have ΣREE of 704–2660 ppm (average 1270 ppm), ΣLREE of 18–212 ppm (average 89 ppm) and ΣHREE of 623–2550 ppm (average 1181 ppm), ΣLREE/ΣHREE of 0.02–0.17, δEu of 0.12–0.38 (average 0.24) and δCe of 1.16–125.04 (average 18.38). On chondrite–normalized REE patterns (Fig. S1), these spots show depletion of LREEs, enrichment of HREEs, and clear negative Eu anomalies and positive Ce anomalies.

Fifteen zircons were analyzed for Lu-Hf isotopes. The zircons have $^{176}Hf/^{177}Hf$ ranging from 0.282295 to 0.282617, which converts to εHf(t) values of 3.72 to -7.83 and two-stage Hf isotope depleted mantle model age ($T_{DM2}$) of 1187 Ma to 1917 Ma by using the weighted mean age.

### 5.2.2 K-feldspar granite

Zircons from K-feldspar granite (JC-20-19) are mostly euhedral, transparent, colorless, and 80–130 μm in length, and have
bright CL intensity and oscillatory zoning (Fig. 6c) with Th/U ratios of 0.29–1.01. Among these, 17 spots cluster on the concordia curve with $^{206}Pb/^{238}U$ ages of 432–446 Ma, defining a weighted mean age of 439.4±2.0 Ma (MSWD=0.88) (Fig. 6c, d). Seven spots have ages of 825 –2344 Ma. In this paper, the $^{206}Pb/^{238}U$ age and $^{207}Pb/^{206}U$ ages are determined for younger zircons (<1000 Ma) and older grains (>1000 Ma). Among them, spot #2, #20 and #22 are weakly luminescent and lacking discernible internal structure with Th/U ratios of 0.03-0.09. They have $^{207}Pb/^{206}Pb$ ages of 1857 Ma, 1847Ma and 1894 Ma,
respectively. Spot #5, #7 and #14 have bright CL intensity and oscillatory zoning with Th/U ratios of 0.48-1.00. They have $^{207}Pb/^{206}Pb$ ages of 2216 Ma, 2344 Ma and 2307 Ma, respectively.

Thirteen zircon spots, which define the weighted mean age of 439.4±2.0 Ma, have ΣREE of 542–1414 ppm (average 862 ppm), ΣLREE of 26–260 ppm (average 80 ppm) and ΣHREE of 502–1154 ppm (average 783 ppm), ΣLREE/ΣHREE of 0.04–0.23, δEu of 0.15–0.58 (average 0.34) and δCe of 1.78–129.07 (average 20.97). All the spots show depletion of LREEs,
enrichment of HREEs, and clear negative Eu anomalies and positive Ce anomalies.

Thirteen zircons were analyzed for Lu-Hf isotopes from sample JC-20-19. Twelve zircon spots, which define the weighted mean age of 439.4±2.0 Ma, show $^{176}Hf/^{177}Hf$ and εHf(t) values of 0.282494 to 0.282056 and -0.49 to -16.27, respectively, with $T_{DM2}$ of 1453 Ma to 2446 Ma. The $^{176}Hf/^{177}Hf$ ratios of #6 (825 Ma) is 0.281812, which converts to εHf(t) value of -16.07, and $T_{DM2}$ of 2722.

## 6 Discussion

### 6.1 Formation age

The Longshoushan is characterized by numerous Early Paleozoic magmatic rocks, which are mainly acidic-intermediate in composition, associated with some mafic rocks. The published age spread is summarised in Fig. 7.



| Age (Ma) | Number and relative probability | Stages of magmatic activities | Main rock assemblage | Geochemical feature and isotopic feature | Genesis type of granite |
|---|---|---|---|---|---|
| | | Stage 3 ca.435-ca.410 Ma | (Monzonitic) granite (Quartz) syenite Granodiorite Diorite Diabase/gabbro | Negative εHf(t) value<br>Subalkaline<br>Metaluminous to peraluminous | I-type and A-type granite<br>Partial melting of crustal materials |
| | | Stage 2 ca.444-ca.435 Ma | Granite Monzonite Granodiorite Diorite Gabbro | Negative εHf(t) value<br>Mainly subalkaline<br>Metaluminous to peraluminous | I-type granite<br>Crust-mantle mixing |
| | N=37 | Stage 1 ca.460-ca.444 Ma | Granite Granodiorite Diabase/gabbro | Positive εHf(t) value<br>Alkali to subalkaline<br>Mainly metaluminous | I-type and A-type granite<br>Partial melting of crustal materials |

**Figure 7 Integrated column of magmatism evolution of Early Paleozoic in the Longshoushan area. On the left is the histogram of age distribution of Early Paleozoic magmatic rocks in the Longshoushan area. Data are in Supplementary materials Table S1**

In this study, zircons from the monzogranite and K-feldspar granite can be divided into two groups. One group (the zircons from the monzogranite and the majority of zircons from the K-feldspar granite) have high Th/U ratio values (0.10-1.27), as well as bright intensity and oscillatory zoning in the CL images. The above characteristics are consistent with those of igneous zircon (Hoskin and Schaltegger, 2003). Hence, the weighted mean ages of 440.8±2.1 Ma (MSWD=1.6) and 439.4±2.0 Ma (MSWD=0.88) are taken to represent the magmatic emplacement age of monzogranite and K-feldspar granite, respectively. Four analyses from sample JC-20-9 have $^{207}$Pb/$^{206}$Pb ages of 825 Ma, 2216 Ma, 2307 Ma and 2344 Ma, which are much older than the emplacement ages of the K-feldspar granite. Hence, these zircons are captured or inherited magmatic zircons. The second zircon group (spots #2, #20 and #22 from JC-20-9) have low Th/U ratio values (0.03-0.09), as well as being weakly luminescent and lacking discernible internal structure in the CL images, which suggests that their ages most likely represent the timing of metamorphism (Hoskin and Schaltegger, 2003).

### 6.2 Crystallization conditions

Temperature, oxygen fugacity, and pressure are important indicators for the crystallization conditions of magmatic rocks. They play a significant role in investigating the origin and evolution of magmas, as well as their relationship with mineralization processes (Xiao et al., 2017; Zeng et al., 2022).



Zircon saturation thermometry ($T_{Zm}$) and the Ti-in-zircon thermometer ($T_{Zr-Ti}$) are commonly used methods for estimating magma temperatures. Based on the calculation method introduced by Watson and Harrison (1983), the $T_{Zm}$ values for the monzogranite and K-feldspar granite are in the range of 839-842 °C (with an average value of 840 °C) and 745-817 °C (with an average value of 782 °C), respectively. The solubility of Ti in zircon is related to the temperature during zircon formation,

and the activities of $TiO_2$ ($aTiO_2$) and $SiO_2$ ($aSiO_2$) in the melt (Watson and Harrison, 2005; Ferry and Watson, 2007). Since both monzogranite and K-feldspar granite contain abundant quartz and zircon, but rare ilmenite and xenotime, the $aSiO_2$ and $aTiO_2$ are 1 and 0.5, respectively (Schiller and Finger, 2019). According to the calculation method proposed by Ferry and Watson (2007), the Ti temperatures of 16 autocrystic zircons from the monzogranite range from 727 to 883 °C (with an average of 791 °C), while the Ti temperatures of 13 autocrystic zircons from the K-feldspar granite range from 687 to 851 °C (with an

average of 756 °C). In addition, the $Al_2O_3/TiO_2$ ratio of the whole rock can be used as an indicator of the formation temperature of granites (Hui et al., 2021). Hot granites typically have lower $Al_2O_3/TiO_2$ ratios, while cold granites have higher $Al_2O_3/TiO_2$ ratios (Sylvester, 1998). The samples of monzogranite have lower $Al_2O_3/TiO_2$ ratios (72.0 to 72.4) than that of the samples of K-feldspar granite (154.5 to 246.6), implying that the melts of monzogranite are hotter than those of K-feldspar granite. This result is in consistent with the outcomes obtained from zircon saturation thermometry ($T_{Zm}$) and Ti-in-zircon thermometer.

Due to the distinct partitioning behavior of $Ce^{4+}$ and $Ce^{3+}$ in zircon, the cerium in zircon can reflect the oxygen fugacity condition of the magmatic systems (Trail et al., 2012). Using the formulas proposed by Trail et al. (2012), the $logfO_2$ for the monzogranite and K-feldspar granite falls within the range of -27 to -10 (with an average of -18) and -26 to -6 (with an average of -19), respectively. In the $logfO_2$-T (°C) and Ce/Ce*-T (K) diagrams (Fig. 8), the monzogranite and the K-feldspar granite mostly fall below the FMQ buffer, indicating low oxygen fugacity conditions.

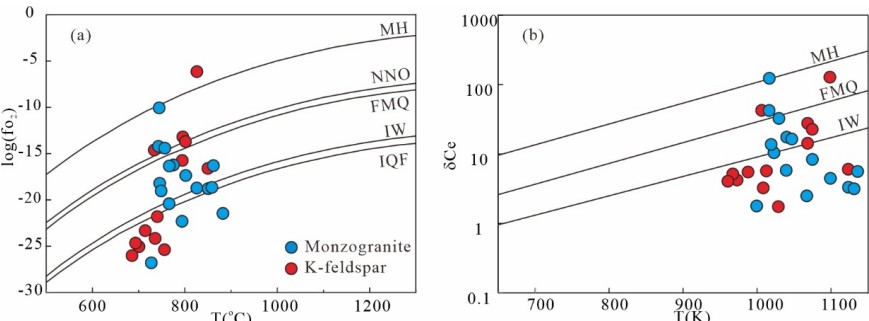

**Figure 8 Diagrams of T-log(f0₂) (a) and T-δCe (b)**

Generally, when garnet is the dominant residual phase in the source region, the melt that forms will exhibit a significant inclined HREE distribution pattern, with Y/Yb > 10 and Ho/Yb > 1.2 characteristics (Tang et al., 2021). In this study, the HREE distribution curves of the monzogranite and K-feldspar granite are relatively flat, with Y/Yb ranging from 7.91 to 10.93,

and Ho/Yb ranging from 0.33 to 0.38, indicating that garnet is not the dominant residual phase. Therefore, the pressure of the



granite source region should be less than 1.7 GPa (mostly less than 1.5 GPa, Tang et al., 2021). The Yb contents of the monzogranite and the K-feldspar granite are both less than 2 ppm (except for one sample), and the Sr contents of the two plutons are both less than 400 ppm, indicating the characteristics of low Yb and low Sr granite (Zhang et al., 2006). In addition, these two plutons have the characteristic low $Al_2O_3$ contents and a distinct negative Eu anomaly. These features are consistent

with the low Sr and low Yb granites in North Hebei, implying that the monzogranite and the K-feldspar granite are partial melts of high-pressure granulites containing plagioclase, clinopyroxene, garnet, and amphibole, at the bottom of the thickened lower crust (Li et al., 2004; Zhang et al., 2006), with source pressures estimated to be between 0.8 and 1.5 GPa (Zhang et al., 2010).

### 6.3 Granitoid petrogenesis

The monzogranite and K-feldspar granite have low Zr (58–175 ppm), Ce (37.6–94.5 ppm), Zr + Nb + Ce + Y (129.6–296.9 ppm) and FeO∗/MgO (4.3– 12.2), distinct from typical A-type granites (Zr>250 ppm, Ce>100 ppm, Zr + Nb + Ce + Y>350 ppm, FeO∗/MgO>16; Whalen et al., 1987). In addition, the magma temperature of the monzogranite ($T_{Zm}$: 839-842 °C; $T_{Zr-Ti}$: 728 to 883 °C) and K-feldspar granite ($T_{Zm}$: 745-817 °C; $T_{Zr-Ti}$: 687 to 851 °C) is significantly lower than that of typical A-type granite (> 900 °C, Skjerlie and Johnston, 1992). Furthermore, all compositions fall in the region of I- and S- type granite

rather than the A-type granite in the discrimination diagrams (Fig. 9a-c). Thus, the monzogranite and K-feldspar granite are not A-type granites. The samples of monzogranite and K-feldspar granite have A/CNK < 1.1, normative corundum <1% in the CIPW norm calculation and there is no presence of peraluminous minerals, such as cordierite, andalusite, muscovite and garnet, which is distinctly different from the mineralogy of S-type granites (Chappell and White, 1992). As shown in Th-Rb and $SiO_2$-$P_2O_5$ diagrams (Fig. 9d and e), the positive correlation between Th and Rb and the negative correlation between $P_2O_5$ and $SiO_2$

also demonstrate the evolutionary trend of typical I-type granite (Wolf and London, 1994). Moreover, In the P-(REE+Y) diagram (Fig. 9f), autocrystic zircons in both the monzogranite and the K-feldspar granite have lower P concentrations, and the concentration of P and REE+Y does not show an obvious positive correlation, showing the characteristics of zircons in I-type granite rather than S-type granite.



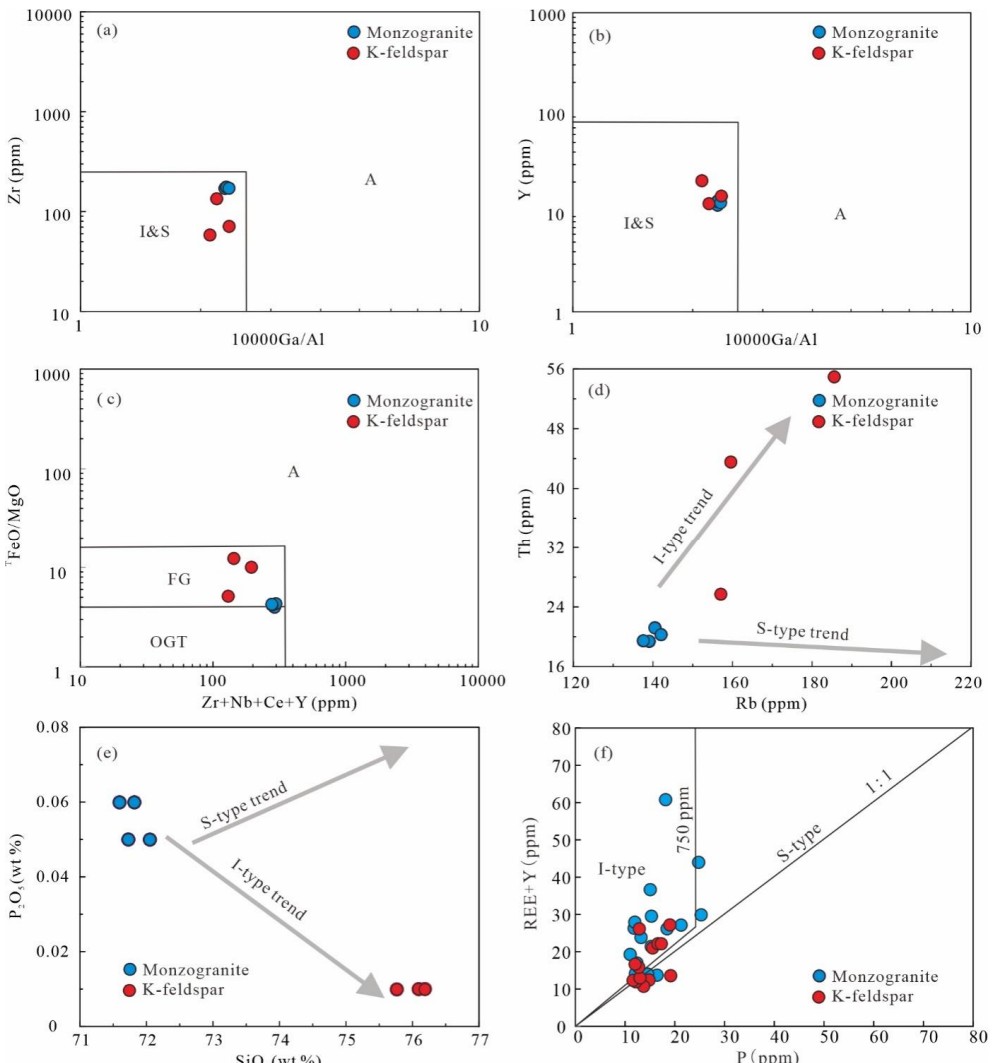

**Figure 9 Discrimination diagram of granitoid genetic types. (a, b, c) 10000 Ga/Al vs. Zr, 10000 Ga/Al vs. Y and Zr+Nb+Ce+Y vs. TFeO/MgO (after Whalen et al., 1987); (d) Rb vs. Th; (e) SiO₂ vs. P₂O₅; (f) P vs. REE+Y (after Burnham and Berry, 2017).**

As noted, the monzogranite and K-feldspar granite have close temporal and spatial relationship, similar $Na_2O+K_2O$ contents (Fig. 4a), uniform REE and trace element patterns (Fig 5c, d). In addition, although there is a difference in $SiO_2$ contents between the two, they show a clear linear relationship in Harker diagrams (Fig. 10). These characteristics suggest a



295       co-magmatic origin. The genesis of highly fractionated I-type granite has been proposed to two primary mechanisms: (1) fractional crystallization (FC) or assimilation fractional crystallization (AFC) of mantle-derived mafic magmas (He et al., 2019), or (2) partial melting of crustal materials with involvement of some mantle-derived components, accompanied by subsequent fractional crystallization (Liu et al., 2021).

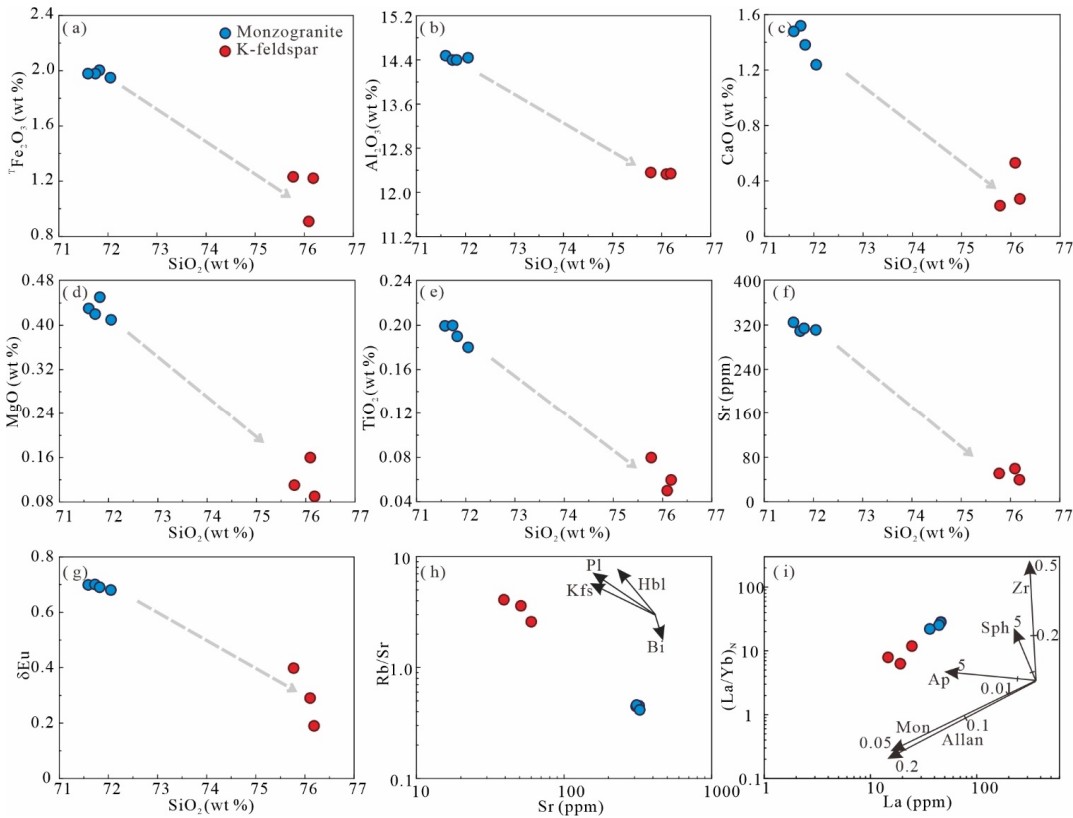

300       **Figure 10. SiO₂ vs. major element oxides (a–e), SiO₂ vs. Sr (f), Sr vs. Rb/Sr (h), Sr vs. Ba(g), and La vs. (La/Yb)$_N$ (i) for the Dakouzi pluton (the partition coefficients of Rb, Sr, and Ba are from Philpotts and Schnetzler, 1970; the partition coefficients of La and Yb are from Ewart and Griffin, 1994)**

**Pl- plagioclase; Kfs- K-feldspar; Bt- biotite; Zr-zircon; Sph- sphene; Ap- apatite; Mon- monazite; Allan- allanite.**

      Mantle-derived primary melt would be expected to have high MgO, TFe₂O₃, CaO and Cr contents (Ma et al., 2013),

which are obviously inconsistent with the characteristics of the monzogranite and K-feldspar granite in this study. Furthermore, mafic melts can only form a minor volume of granitic melts through crystal differentiation, with a proportion of about 9:1 (Zeng et al., 2016). However, in the Longshoushan, most of the ~440 Ma magmatic rocks are granitic rocks, and no large-



scale exposures of mafic-ultramafic rocks have been reported. Therefore, monzogranite and K-feldspar granite are unlikely to have been formed by the FC or AFC processes of mantle-derived magma.

The Hf isotopes of zircon are not affected by fractional crystallization, and thus can provide precise constraints on the origin of magma (Griffin et al., 2002). The εHf(t) values of the monzogranite range from -0.37 to -16.28, while those of the K-feldspar granite range from 3.53 to -7.74, both exhibiting a large range of variation in εHf(t) values (Fig. 11). This phenomenon indicates the existence of different Hf isotopic end-members in the source regions of these two plutons, which could be attributed to the mixing of mafic and felsic melts (Zhang et al., 2015; He et al., 2019), heterogeneous composition of

source crust materials (Zeng et al., 2016), or variable degrees of melting of source minerals (Zeng et al., 2021). In addition, some quartz and biotite boundaries have been melted and transformed into irregular embayments, and plagioclase inclusions occur within the quartz. These observations suggest the injection of high-temperature magma during the magma evolution process (Duan et al., 2021). On the Rb/V-1/V and La/Cr-1/Cr diagrams (Fig. 12a and b), samples from monzogranite and K-feldspar granite exhibit a linear distribution, indicating magma mixing (Liu et al., 2021). On the MgO-FeO$_T$ and SiO$_2$/MgO-

Al$_2$O$_3$/MgO diagrams (Fig. 12c and d), our samples also show an obvious magma mingling trend (Tang et al., 2021). In addition, Zhang et al. (2021) reported an age of ~441 Ma for the Xijing clinopyroxene diorite vein, and Wang et al. (2019) reported crystallization ages of ~440 Ma for the mantle-derived mafic microgranular enclaves in the Jiling granite. Therefore, there was mantle-derived magmatism in the Longshoushan at ~440 Ma, which could provide the mantle material for the granitic magmatism. We use the ~441 Ma Xijing clinopyroxene diorite (Zhang et al., 2021) and our sample of K-feldspar granite (JC-

20-20) to assess magma mixing simply in terms of two end-member compositions. Our simulation results show a curvilinear evolutionary relationship on the Th/Nd-Th and Rb/Nd-Rb diagrams (Fig. 12e and f), which is also consistent with magma mixing (Schiano et al., 2010; Tang et al., 2021). To conclude, the monzogranite and K-feldspar granite in this study were derived by magma mixing of mantle-derived and crust-derived magmas.



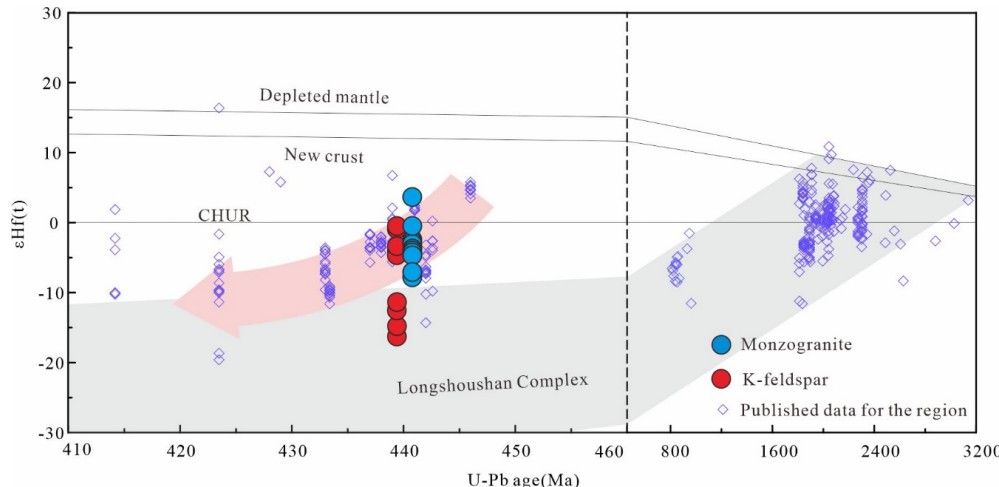


**Figure 11 Zircon εHf(t)-age (Ma) diagram for samples in this study and published data for the region. The source of the published data can be found in Supplementary materials Table S1.**



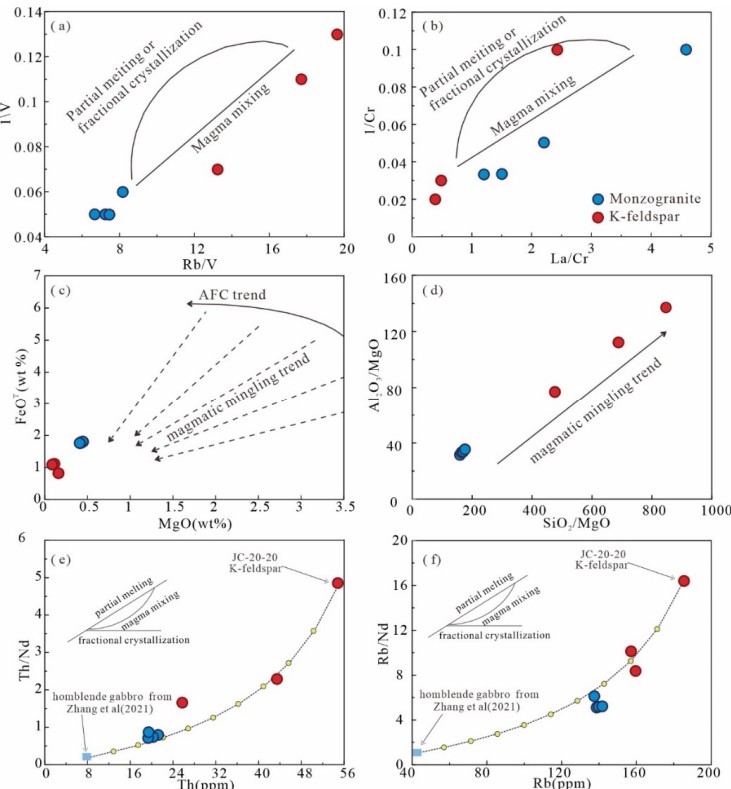

**Figure 12 Diagram of magma mixing discrimination. (a) Rb/V vs. 1/V diagram; (b) La/Cr vs. 1/Cr diagram; (c) MgO vs. FeO$^T$ diagram; (d) SiO$_2$/MgO vs. Al$_2$O$_3$/MgO diagram; (e) Th vs. Th/Nd diagram; (f) Rb vs. Rb/Nd diagram (a and b after Schiano et al., 2010; e after Zorpi et al, 1989)**

As shown in Fig. 11, some spots from both rock bodies fall within the evolutionary trend line of the Early Precambrian basement strata, known as the Longshoushan Complex. Furthermore, abundant 1.85-1.90 Ga metamorphic zircons and ~2.3 Ga plutons have been identified in the Longshoushan Complex (Gong et al., 2016; Zeng et al., 2018), which corresponds to the inherited zircon ages found in the K-feldspar granite. Therefore, the Longshoushan Complex is mostly likely the crustal source of the granitoids in this study. Compared with the K-feldspar granite, the monzogranite has higher εHf(t) values and is closer to the mantle end-member in the mixing diagrams, indicating a relatively lower proportion of crust-derived magmas.

All the samples of the monzogranite and the K-feldspar granite are characterized by relatively high SiO$_2$ and total alkali contents, low TFe$_2$O$_3$, MgO and CaO contents, high differentiation index, and enrichment of Rb, Th and U, indicating that these rocks are highly fractionated (Xiao et al., 2014). In this study, the granitoids have negative Eu-Ba-Sr anomalies (Fig. 5c and d), indicating plagioclase and K-feldspar fractionation (Harris et al., 1990). On the Rb/Sr vs. Sr diagrams (Fig. 10g), the



granitoid samples show a separation trend of plagioclase. The negative correlation of $SiO_2$ with TFeO and MgO suggests fractionation of amphibole and biotite during magmatic evolution (Fig. 10a and d). As shown in the Sr-Ba diagram (Fig. 10h), amphibole is likely to dominate the fractional phase. The increasing $TiO_2$ content with decreasing $SiO_2$ (Fig. 10e), along with

the clear Ti, Nb and Ta anomalies in the primitive-mantle-normalized trace element diagrams (Fig. 5b), implies the fractionation of rutile (Foley et al., 2000). In a diagram of $(La/Yb)_N$ vs. La (Fig. 10i), the variation of REE contents suggests the fractionation of monazite and allanite. Hence, the monzogranite and K-feldspar granite have experienced extensive fractional crystallization of plagioclase, K-feldspar, amphibole, biotite, rutile, monazite and allanite.

### 6.4 Tectonic implications

#### 6.4.1 Tectonic affinity of the Longshoushan in the Paleozoic

During the Paleozoic to Mesozoic, the Alxa Block was situated between multiple tectonic domains, with its northern and southwestern sides experiencing subduction-accretionary orogenesis of the Central Asian Orogenic Belt and the North Qilian Orogenic Belt, respectively (Song et al., 2013; Liu et al., 2017; Xue et al., 2017; Zhang and Gong et al., 2018; Hui et al., 2021). The North Qilian Orogenic Belt is a component of the much larger Central China Orogenic Belt (CCOB), which includes areas

~2000 km to both the east and west of the Alxa Block.

It is widely believed that the Yabulai-Nuoergong-Honggueryulin zone and Zongnaishan-Shalazhashan zone at the northern side of the Alxa Block were mainly influenced by the Central Asian Orogenic Belt (Liu et al., 2017; Zhang and Gong et al., 2018; Hui et al., 2021). As shown in Fig.13 a, magmatic activities in these areas were mainly concentrated in the late Paleozoic-Mesozoic, with a peak age of ~270 Ma. The magmatic activity in the Longshoushan, which lies at the southern

margin of the Alxa Block, mainly occurred in the Early Paleozoic with a peak age of ~440 Ma (Fig. 13d), which is significantly different from the magmatic events in the northern side of the Alxa Block and basically consistent with the Hexi Corridor, North Qilian Orogenic Belt and Central Qilian Block (Fig. 13e-g), as well as the wider CCOB (Allen et al., 2023). Therefore, the Longshoushan was most likely part of the evolution of the CCOB during the Paleozoic, and specifically adjacent regions of the North Qilian Orogenic Belt and the Hexi Corridor. There is a small amount of Late Paleozoic magmatic rocks in the

Longshoushan, such as the Taohualashan pluton (Xue et al., 2017). These plutons are mainly distributed on the northwest side of the Longshoushan (Fig. 1b), close to the northern boundary of the Alxa Block (near the Central Asian Orogenic Belt), and are considered to related to the Central Asian Orogenic Belt (Xue et al., 2017; Zhang et al., 2021). The Beidashan is located to the north of the Longshoushan, and both Early Paleozoic and Late Paleozoic-Mesozoic magmatic rocks are extensively exposed there (Fig. 13c). This feature indicates that the Beidashan is located at the intersection of two tectonic domains, and

was influenced by both the Central Asian Orogenic Belt and the North Qilian Orogenic Belt. In addition, similar to the Longshoushan, the Early Paleozoic magmatic rocks in the Beidashan are mainly distributed on the southeastern side, while the Paleozoic-Mesozoic magmatic rocks are mainly distributed on the northwest side (Fig. 1b). Therefore, in the Alxa Block, the Longshoushan and the southeast side of the Beidashan is mainly influenced by the North Qilian Orogenic Belt, while other



areas to the north are mainly influenced by the Central Asian Orogenic Belt. Apart from Beidashan and Longshoushan, Early

Paleozoic magmatic rocks are mainly distributed in the eastern part of the Alxa Block, including Bayan Nuru, Bayanwulashan, and Langshan areas. Some scholars suggested that these magmatic events are related to the convergence between the Alxa Block and the North China Craton (Dan et al., 2016; Wang et al., 2015).

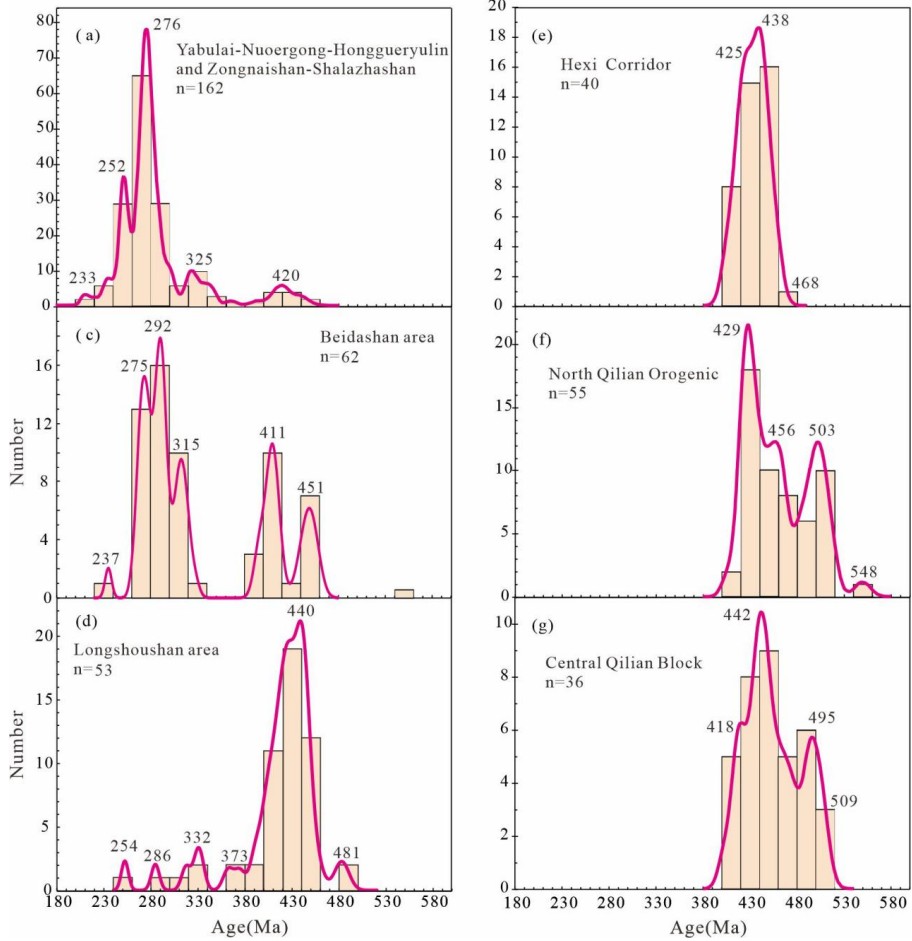

**Figure 13 Kernel density estimate (KDE) plots for Palaeozoic and Mesozoic magmatic ages in the Alxa Block, North Qilian Belt**
**and Central Qilian Block. Ophiolitic rocks are not included. Data are in Supplementary materials Table S1. All dates are zircon U-Pb ages.**

Figure 13 indicates that magmatism in the Alxa Block started later than the >500 Ma age common in other parts of the

CCOB to the south, and the region lacks the twin peaks in Early Paleozoic ages for magmatism found in other areas such as



the North Qilian Orogenic Belt and Central Qilian Block (Allen et al., 2023). We suggest that the reason for this difference in

age spectrum is that the Longshoushan was initially far enough away from the Proto-Tethyan subduction zone that it was not affected by arc magmatism, which was focused on the North Qilian Orogenic Belt and Central Qilian Block to the south. Detailed discussion can be found in Section 6.4.3.

### 6.4.2 Evaluation of crustal thickness

Hu et al. (2017) designed an empirical formula between the median Sr/Y ratio of intermediate-acid rocks and the crustal

thickness in continental collisional orogens. In addition, Mantle and Collins (2008) designed the empirical relationship between the maximum Cr/Y ratio of basalt and crustal thickness. Therefore, Sr/Y and Ce/Y ratios of magmatic rocks can be used as proxies for crustal thickness. The Longshoushan experienced the convergence and collision of the Alxa Block and the basement of the North Qilian Orogenic Belt in the Early Paleozoic, so the intermediate-acid rocks and basic-ultrabasic rocks formed in this process can be used to estimate the thickness of the paleo-crust by using the formulas designed by Hu et al. (2017) and

Mantle and Collins (2008), respectively.

In this study, we collected published data for 153 magmatic rock samples of Early Paleozoic age in the Longshoushan, including 8 ultrabasic rock samples, 31 basic rock samples, 45 intermediate rock samples and 88 acidic rock samples. According to the location, lithology and formation age of these rocks, the above samples are divided into 31 subsets in this study. Among them, 26 subsets of intermediate-acid rocks were run through the data processing scheme recommended by Hu

et al., (2017) to process data. The first step is to eliminate the samples with $SiO_2>72\%$ and $MgO<0.5\%$ or $>6.0\%$ in each subset; The second step is to remove individual data points with Sr/Y outliers from each data subset using the modified Thompson tau statistical method. The third step is to eliminate the subset whose average Rb/Sr value is greater than 0.35. After the above process, there are 6 subsets of 32 samples left. According to the calculation formula of Hu et al., (2017), the crustal thickness is calculated by the median ratio of Sr/Y in each subset of samples. On the other hand, 5 subsets of basic-ultrabasic rock

samples were processed by the Mantle and Collins (2008) method. The first step is to eliminate the samples with $SiO_2<44\%$, $MgO<4\%$ and $LOI>4\%$. In the second step, samples of alkaline and shoshonitic basalts derived from the enriched lithospheric mantle are excluded. After the above process, there are 4 groups of 28 samples left. According to the calculation formula of Mantle and Collins (2008), the corresponding crustal thickness is obtained through the maximum Ce/Y ratio in each subset. See Table S6 the calculation results. As shown in Fig. 14, the thickness of the crust underwent a transition from thickening to

thinning during the period of 460-410 Ma, with the initial timing of the transition being ~435 Ma.




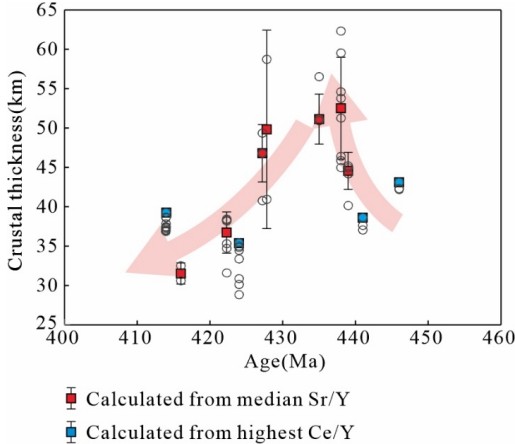

**Figure 14 Plot of crustal thickness versus age of magmatic rocks for each data subset in the Longshoushan, southwestern Alxa Block. Data are in Supplementary materials Table S1.**

**6.4.3 Tectonic evolution of the Longshoushan in the Early Paleozoic**

A notable feature of the Longshoushan is the large volume of Late Ordovician-Silurian magmatic rocks. Based on the statistics and analysis of U-Pb ages and geochemical data of these rocks, three stages of magmatic activities can be divided (Fig. 7).

The first stage (460 Ma–444 Ma): The magmatic rocks in this period are all acidic rocks (except one mafic rock, Fig. 7), calc-alkaline (alkali-calcic) granites (Fig. 4), which is consistent with the magmatic rock assemblage formed in a subduction environment (Song et al., 2015; Wang et al., 2020). Additionally, the 461–444 Ma magmatic rocks are enriched in LREEs and

LILEs, depleted in HREEs and HFSEs, and exhibit significant negative anomalies of Nb, Ta, P and Ti (Fig. 5a and b), which are consistent with the characteristics of arc-derived magmas (Jahn et al., 1999; Li et al., 2016). Hence, the magmatic events during this period are related to slab subduction. During this period, there are some A-type granites, which are possibly related to the extension caused by the rollback of the subducting slab. This phenomenon has been observed in the Central Asian Orogenic Belt as well as in eastern Australia (Cawood et al., 2011; Wu et al., 2021).

In the North Qilian Orogenic Belt, there are two ophiolite belts, the Northern Ophiolite Belt (NOB) and the Southern Ophiolite Belt (SOB), separated by the North Qilian Volcanic Arc (NQVA). Allen et al. (2023) named the oceans represented by these two ophiolite belts as the main Proto-Tethys ocean and North Qilian back-arc basin (NQ bab), respectively. The Dachadaban tholeiite-boninite suite (517-487 Ma) and Laohushan IAB-type boninite (492-488 Ma) are considered to have been formed by the northward subduction of the Proto-Tethys on the south side of the NQVA (Xia et al., 2012; Fu et al., 2022).

Their formation ages indicate that this subduction was already underway in the mid-late Cambrian (Fig. 15a). Under this tectonic background, intense magmatic activity occurred in the NQVA and SOB during the mid-late Cambrian and early Ordovician. In comparison, the initiation age of the northward subduction of the NQ bab is later. The formation age of SSZ-



type ophiolites (gabbro, basalt, and plagiogranite) formed in the northward subduction of NQ bab is 454-448 Ma (Fu et al., 2020), while the age of metamorphism of the Laohushan boninite related to this subduction event is ~455 Ma (Fu et al., 2022).

Therefore, there is no intense Cambrian magmatic activity in the NOB, Hexi Corridor (HC), and Longshoushan. The magmatic activity in these areas gradually intensified in the Middle-Late Ordovician, which is consistent with the NQ bab undergoing northward subduction by this time (Fig. 15b).

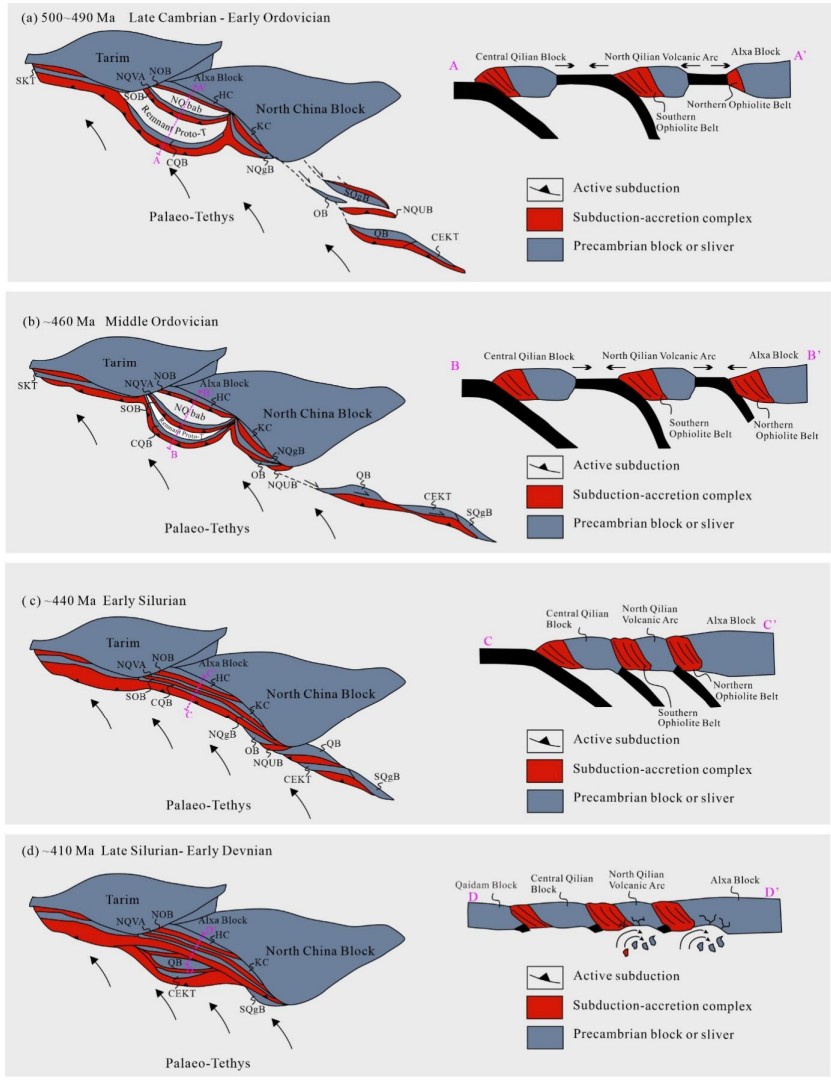





**Figure 15 Tectonic model illustrating the four stages of evolution of the CCOB.**

**Abbreviations are as follows. CEKT: Central East Kunlun Terrane. CQB: Central Qilian Block. HC: Hexi Corridor. KC:
Kuangping Complex. NOB: Northern Ophiolite Belt. NQgB: North Qinling Belt. NQUB: North Qaidam Ultra High-pressure Belt.
NQVA: North Qilian Volcanic Arc. OB: Oulongbuluke Block. SKT: South Kunlun Terrane. SOB: Southern Ophiolite Belt. SQgB:
South Qinling Block. QB: Qaidam Block.**

The second stage (444–435 Ma): Compared to the first stage of magmatic activity, the intensity of the 444-435 Ma
magmatic activity significantly increased, and the rocks have higher total alkali content (Fig. 4a and b). In this stage, although
calc-alkaline granite was still dominant, there are also some alkaline rocks and the presence of mantle-derived magmatic
activity (Fig. 7). The above magmatic rock assemblage is consistent with the post-collisional magmatic rock assemblage
formed after a major ocean closed (Zhang et al., 2019; Wang et al., 2020). Noting that the meaning of post-collisional has

multiple interpretations, the post-collisional stage refers to the stage after the initial collision between continental plates or
between continental plates and island arcs in this study (Liegeois et al., 1998). Furthermore, the Hf isotope characteristics
indicate a significant decrease in zircon εHf(t) during the 444-435 Ma period compared to the previous stage (Fig. 11),
reflecting the increasing contribution of ancient continental crust components in the magma source region. This phenomenon
is also consistent with the variations in magmatic rocks during the transition from subduction to collision (Li et al., 2023). As

shown in Fig. 14, significant crustal thickening occurred in the Longshoushan during the period of 444-437 Ma. In addition,
research has shown that thickened crust also occurred in the HC, NOB and NQVA on the south side of the Longshoushan
during 450-430 Ma (Zhang et al., 2017; Yang et al., 2019). The above phenomena are consistent with the shortening and
thickening of continental crust caused by the closure of ocean basins and subsequent continent-continent collision (Li et al.,
2023). In addition, the latest arc-type magmatic rocks in the North Qilian Orogenic Belt were found to have formed at 446 Ma

(Wang et al., 2005). Silurian molasse rocks are widely distributed and unconformably overlie pre-Silurian strata in this
orogenic belt (Song et al., 2013; Xia et al., 2016; Zeng et al., 2021). Hence, it is generally believed that the closure of the NQ
bab occurred at the end of the Ordovician (~444 Ma, Song et al., 2013; Zeng et al., 2021). Therefore, the Longshoushan
transitioned from a subduction environment to a post-collisional environment from ~444 Ma (Fig. 15c).

Collectively, the characteristics of the Longshoushan rocks from 444-435 Ma are similar to the signatures of the

magmatism after 440 Ma in the North Qaidam Ultra High-Pressure Belt (NQUB) (Allen et al, 2023), presently ~500 km to the
southwest of the Longshoushan. This coincidence in timing and evolution requires either a chance juxtaposition of separate
arcs and continents across separate subduction zones, or, and more likely in our view, the duplication of sections of what was
once a single subduction and collision system.

The third stage (435–410 Ma): During the evolution of orogenic events, calc-alkaline granites are commonly replaced

gradually by high-silica alkaline to peralkaline magmatic rocks (Fig. 4), representing the transition from the post-collisional
environment to the intraplate extensional environment (Liégeois et al., 1998; Bonin, 2004). In the Longshoushan, the 444-435
Ma magmatic rocks are mainly high potassium calc-alkaline I-type granite (Fig. 7). However, the proportion of alkaline rock
series significantly increased and multiple A-type granites appeared during 435–410 Ma, indicating the characteristics of an



intracontinental extensional environment. During this period, a significant number of mantle-derived mafic dykes were
developed, which also reflects an extensional setting (Zeng et al., 2021). Therefore, the Longshoushan transitioned from a
post-collisional compressional environment to an intraplate extensional setting at ~435 Ma. As discussed earlier, the transition
in crustal thickness from thickening to thinning in the Longshoushan also occurred at ~435 Ma, which is consistent with the
inferred stress changes in the study area (although we cannot rule out efficient erosion as the cause of crustal thinning). The
driver for the proposed extension is not clear, but it would have been enhanced by a crust with initially greater thickness that
normal.

There are two different views regarding the mechanism of the transition from a compressional environment to an
extensional environment in the North Qilian orogenic belt: slab break-off (Xia et al., 2016; Zeng et al., 2021) and lithospheric
delamination (Yu et al., 2015; Zhang et al., 2017). Studies have shown that the mantle-derived magmas in this stage mainly
derived from the enriched lithospheric mantle, indicating that the lithospheric mantle in the Longshoushan has not been
completely delaminated (Zeng et al., 2021). Small-scale delamination of hydrated lithosphere is a possible generation
mechanism (Kaisalaniemi et al., 2014) (Fig. 15d). A complementary mechanism is adiabatic melting during extension and
lithospheric thinning, supported by the observations of crustal thinning over time. These mechanisms are not mutually
exclusive. While we cannot fully rule out slab break-off, we note modelling studies that suggest it does not normally generate
significant magmatism (Freeburn et al., 2017).

**7 Conclusions**

(1) LA-ICP-MS zircon U-Pb dating indicates that the monzogranite and the K-feldspar granite in the east of Longshoushan
formed at 440.8±2.1 Ma (MSWD=1.6) and 439.4±2.0 Ma (MSWD=0.88), respectively.

(2) The monzogranite and K-feldspar granite in the east of the Longshoushan were generated through various degrees of
mixing between crustal-derived magma from the thickened lower crust and mantle-derived magma, and then underwent
extensive fractional crystallization with removal of plagioclase, K-feldspar, amphibole, biotite, rutile, monazite and allanite.

(3) Based on the temporal and spatial distribution of the magmatic rocks in the Longshoushan, we propose a three-stage
tectonic evolution model. ① 461 to 445 Ma: The northward subduction of NQ bab resulted in the develop of arc-type magmatic
rocks. ② 444 to 435 Ma: The closure of the NQ bab and subsequent continent-continent collision resulted in significant crustal
thickening in the study area. ③ 435 to 410 Ma: Tectonics transitioned from a collisional compressional environment to an
intraplate extensional environment, evidenced by crustal thinning at this time.

*Data availability*. The data presented in the supplement.

*Supplement*. The supplement related to this article (Tables S1, S2, S3, S4, S5 and S6) is available online at:



*Author contributions.* Renyu Zeng: Conceptualization, Methodology, Writing - original draft. Hui Su: Writing- original draft,
visualization. Mark B. Allen: Writing - review & editing; Haiyan Shi: Visualization; Chenguang Zhang: Visualization; Jie
Yan: Funding acquisition.

*Competing interests.* The contact author has declared that neither they nor their co-authors have any competing interests

### Acknowledgements

We thank Dr. Zhijie Song and Dr. Liangpeng Deng for constructive reviews and useful suggestions. We are also grateful to
Qixing Ai and Yuhua Wang for their help with the field work. This research was funded by the Jiangxi Provincial Natural
Science Foundation (No. 20232BAB213061), the China Uranium Industry Co. LTD. - East China University of Technology
Innovation Partnership Foundation (2023NRE-LH-12), National Nature Science Foundation of China (Grants No. 42030809,
42262017, 42002095; 42162013), China Scholarship Council (No. 202008360018), Open Research Fund Program of Key
Laboratory of Metallogenic Prediction of Nonferrous Metals and Geological Environment Monitoring (Central South
University), Ministry of Education (2022YSJS10)

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
