# Peer review of "Petrogenesis of Early Paleozoic I-type granitoids in the Longshoushan and implications for the tectonic affinity and evolution of the southwestern Alxa Block"

_EGUsphere, 2024_

## Author Comment (AC1)

Dear Jan Wijbrans:

Thank you for your comments concerning our manuscript "Petrogenesis of Early Paleozoic I-type granitoids in the Longshoushan and implications for the tectonic affinity and evolution of the southwestern Alxa Block" (EGUSPHERE-2024-1145). Those comments are all valuable and very helpful for revising and improving our paper. We have studied the comments and suggestions carefully and have made corrections. We hope our revisions meet with your approval. Below, the comments are addressed point by point.

1) The authors come up with a three stage tectonic model which is based on a proposed division in magmatism presented in section 6.1 and summarized in figure 7: with first, second and third stages respectively from 460-444 Ma, from 444-435 Ma, and from 435-410 Ma. This division is not further discussed in section 6.1. This discussion only comes in section 6.4.

Considering the Reviewer's suggestion, we have moved the content of the three-stage tectonic model and Figure 7 to section 6.4.3. In section 6.1, we only explain the zircon ages of the two granites involved in this study

2) The discussion in section 6.4 is based exclusively on the published data set. This part of the manuscript is not well integrated with the preceding study of the two granites and their merits for the tectonic interpretation. One would have expected that the outcomes of the geochemical study described in the first part of the manuscript would have been mentioned in section 6.4, where the second stage of the model, the period from 444-435 Ma is discussed in lines 455-478. Given the fact that the two granites that were studied in detail both had emplacement ages around 440 Ma, their geochemical make up as discussed in sections 6.1-6.3,to me seems relevant for the discussion in section 6.4. There is however in section 6.4 no mention of the preceding granite study. And I feel that this is a substantial shortcoming of this discussion section. A better integration of the outcomes of the geochemical study of the two granites in the discussion of the tectonics in section 6.4. is currently missing.

Thank you for your comment. In the discussion of the second stage in section 6.4.3, we have included the outcomes of the geochemical study of the two granites.

The details are as follows:

*The second stage (444–435 Ma): Compared to the first stage of magmatic activity, the intensity of the 444-435 Ma magmatic activity significantly increased, and the rocks have higher total alkali content (Fig. 4a and b).* *In this stage, although calc-alkaline granitic magmatism was still dominant, there are also some alkaline rocks. In addition, as previously mentioned, the ~440 Ma monzogranite and ~439 Ma K-feldspar granite were formed by crust and mantle-derived magma mixing. During this period, mantle-derived intermediate-basic rocks also existed in the Longshoushan (Figure 7). Therefore, there was mantle-derived magmatic activity in the Longshoushan from 444 to 435 Ma.* *The above magmatic rock assemblage is consistent with the post-collisional magmatic rock assemblage formed after a major ocean closed (Zhang et al., 2019; Wang et al., 2020). Noting that the meaning of post-collisional has multiple interpretations, the post-collisional stage refers to the stage after the initial collision between continental plates or between continental plates and island arcs in this study (Liegeois et al., 1998).* *In this study, the zircon εHf(t) values of the monzogranite and K-feldspar granite are primarily negative, with a minimum value reaching -16.27. As shown in Fig. 10,* *the Hf isotope characteristics indicate a significant decrease in zircon εHf(t) during the 444-435 Ma period compared to the previous stage (Fig. 10), reflecting the increasing contribution of ancient continental crust components in the magma source region. This phenomenon is also consistent with the variations in magmatic rocks during the transition from subduction to collision (Li et al., 2023). As shown in Fig. 13, significant crustal thickening occurred in the Longshoushan during the period of 444-437 Ma.* *This is consistent with that the monzogranite and K-feldspar granite in this study were formed through partial melting at the bottom of the thickened lower crust, with source pressures approximately between 0.8 and 1.5 GPa.*

3) To my mind the discussion section, and in particular the sections 6.1 and 6.4 could

do with a substatial revision, including a motivation for the model division into three stages of tectonic and magmatic evolution to section 6.1, and integrating the outcomes of the discussion of section 6.1 to 6.3 better into section 6.4.

Thank you for your comment. We have moved the content of the three-stage tectonic model and Figure 7 to section 6.4.3, as some outcomes from sections 6.2, 6.3, 6.4.1, and 6.4.2 serve as evidence for establishing the tectonic-magmatic evolution model. We have incorporated the outcomes of the geochemical study of the two granites into section 6.4, as detailed in the previous response.

**Technical comments**

4) Figure 1. In the inset 'a' the border between the Alxa block and the North China craton is a Noth-south structure. This North South structure is not shown and cannot be inferred from the detailed map 'b', where it seems more appropriate to inferre a NE-SW trending contact between the two units.

We are very sorry for this mistake. There is considerable controversy regarding the eastern boundary of the Alxa Block, with differing viewpoints including the Helanshan western fault, the western fault of the Bayan Ulanshan, and the Ordos western edge fault on the eastern side of the Helanshan. There are discrepancies in the boundaries selected in Figure 1 a and b. We have redrawn the boundary in Figure 1a to match that in Figure b.

[Figure]

5) Figure 2: inset 'b' is a blow-up of part of overview map 'a', but the location of 'b' on 'a' is not well indicated. Please make the link between the two maps more noticeable.

Thank you for your comment. In Figure a, we have highlighted the area corresponding to Figure b with a bold blue box.

[Figure]

3)Line 139-140: wavelength dispersive XRF. Isn't XRF not always wavelength dispersive.

We have removed 'wavelength dispersive'.

4)Line 162-163. You claim that the samples are in the Calcalkaline series in figure 4b. It seems to me that your data points are 'straddling' the Calc-alkalic – alkali-calcic boundary.

We apologize for the inaccuracy in our description. The revised content is as follows:

*All samples fall in the granite area in the TAS classification (Fig.4a), and the Alkali -calcic to Calc-alkaline series areas in the $SiO_2$-$N_2O$+$K_2O$ diagram (Fig.4b).*

5)Line 179-187: what you don't discuss in this section is the data in fig 5e:the series samples that have no LREE enrichment, which show flat patterns and in a couple of cases even a positive Eu*/Eu peak. Are these really granites? I'd expect such patterns more for diorites or gabbro's.

Thank you for your comment. We apologize for limiting the samples to granite in the figure title. In fact, some data in this figure are from intermediate and mafic rocks, and as you mentioned, the flat patterns represents gabbro samples. Therefore, we have revised the figure title:

*Figure 5 Chondrite-normalized REE patterns and primitive mantle-normalized trace element patterns for Early Paleozoic magmatic rocks (chondrite and primitive mantle values are from Sun and McDonough, 1989). The source of the published data can be found in Supplementary materials Table S1.*

6) Fig 6 b and d, it may be a suggestion to plot the data with the ages going from low to high? Isn't going to change the outcome of the discussion, but it would make evaluation of the data by eyeballing somewhat simpler.

Thank you for your comment. We have revised Figure 6.

[Figure]

7) Line 277: 'Atype' to read 'A-type'

    Revised.

8) Line 282: 'SiO2P2O5 ' to read ' SiO2-P2O5'.

    Revised.

9) Line 286 'Itype' to read 'I-type'

    Revised.

---

## Author Comment (AC2)

Dear Huan Li

Thank you for your comments concerning our manuscript "Petrogenesis of Early Paleozoic I-type granitoids in the Longshoushan and implications for the tectonic affinity and evolution of the southwestern Alxa Block" (EGUSPHERE-2024-1145). Those comments are all valuable and very helpful for revising and improving our paper. We have studied the comments and suggestions carefully and have made corrections. We hope our revisions meet with your approval. Below, the comments are addressed point by point and the revisions are indicated. However, I recommend you read the PDF attached (Supplement), as it will be more convenient for reading and contains images.

**The major comments for modification are outlined below.**

1) Line23: It is no need to display εHf(t) values to two decimal places.

Thank you for pointing this out. We have modified the values of εHf(t) to one decimal place.

2) Line59: The timing of the amalgamation of the Alxa block and North China Craton is in dispute, not unclear. The relationship between the Alxa block and North China have also been discussed by Yuan Wei and Yang zhenyu, 2015 and Li jinyi et al., 2012.

Considering the reviewer's suggestion, we have revised the sentence and added the reference. The revised content is as follows:

*The eastern margin may have undergone collision and amalgamation events with the North China Craton, although the timing of the amalgamation is in dispute (Li et al., 2012; Wang et al., 2015; Yuan et al., 2015; Dan et al., 2016).*

3) There are several instances of improper citations, and many important references are missing. Such as line 58, papers by scholars such as Wang Tao and Wu Fuyuan can be referred to; Line 69-73, the authors neglected earlier researchers, who first to make these points, e.g. Shi Xingjun et al., 2014, and Zhang Lei et al., 2023, and these

work seems to provide some new ideas for the study of magmatism in the Longshoushan Mountains; Line 78-87, When it comes to the Precambrian magmatism of the Alxa block, the findings of some authors should be considered, e.g. Dan Wei ,2012, Wang zengzhen , 2019, Dong chunyan ,2007.

Considering the reviewer's suggestion, we have added these reference:

*The tectono-thermal events in the Alxa Block mainly occurred during the Paleoproterozoic and Paleozoic to early Mesozoic. During the Paleoproterozoic, the Alxa Block experienced the ~2.5 Ga magmatic-metamorphic event, ~2.3 Ga and 2.05-2.0 Ga magmatic events, as well as the 1.95-1.80 Ga magmatic-metamorphic event (Zhang et al., 2013; Gong et al., 2016; Zeng et al., 2018; Qi et al., 2019; Wang et al., 2019).*

*As an important component of the continental crust, granitoid is of great significance in studying crustal properties, tectonic framework and tectonic evolution (e.g. Pearce et al., 1984; Wang et al., 2017; Wu et al., 2017; Zeng et al., 2022).*

*Hence, recent studies have indicated that the Qagan Qulu Ophiolite Belt was most likely a Late Paleozoic suture related to the closure of a back-arc basin, representing the tectonic boundary between the Alxa Block and the Central Asian Orogenic Belt (Shi et al., 2014; Zhang et al., 2015; Hui et al., 2021; Zhang et al., 2023).*

*The ~2.3 Ga and 2.05-2.0 Ga magmatic events are primarily found in the Bayanwulashan, Diebusuge and Longshoushan areas, and they are generally believed to be related to an extensional tectonic setting (Dong et al., 2007; Dan et al.,2012; Zeng et al., 2018). The 1.95-1.80 Ga metamorphic events are widely documented in the metamorphic basement throughout the Alxa Block (Zhang et al., 2013; Gong et al., 2016; Zeng et al., 2018).*

4) Line 110: The size and shape of the pluton have not been described.

Considering the Reviewer's suggestion, we added a sentence to describe the exposed area of the rock body.

*These two plutons occupy an area of approximately 12 km².*

5) Line 343-345: The geochemical identification of high- fractionated granites relies primarily on trace elements, e.g. Cr, Ni, Co, Sr, Ba, Zr and ratio of Zr/Hf、Nb/Ta and Y/Ho.

Thank you for your valuable comment. We added content regarding the degree of differentiation through trace elements.

*All the samples of the monzogranite and the K-feldspar granite are characterized by relatively high $SiO_2$, total alkali and differentiation index, low $TFe_2O_3$, MgO and CaO contents, low Nb/Ta (9.5–14.5) and Zr/Hf values (26.3–40.2), and enrichment of Rb, Th and U, indicating that these rocks are highly fractionated (Xiao et al., 2014; Wu et al., 2017). This is consistent with the samples of these two granites being in the fractionated felsic granites (FG) area in the $FeO^T/MgO$ vs. Zr+Nb+Ce+Y diagram (Fig. 8c).*

6) Figure1: The mafic rocks in northern Alxa region are mostly gabbroic not ultramafic (Fig. 1b), and the research area is not clear in Fig1b.

Thank you for your comment. We marked the location of the study area in Figure 1b with a red pentagram. The study area is primarily composed of gabbro, but there are also small amounts of ultramafic rocks, which is why there is a legend for ultramafic rocks.

[Figure]

7) Line 401-408: The authors removed a large amount of raw data based on Hu's methodology, can you explain the rationale?

The criteria for removing data in this paper are based on the paper by Hu and Mantle. The reasons include exceeding the range of original data that fits the formula, the presence of outliers, and samples that are strongly influenced by differentiation. We have added content in the MS explaining the reasons for data removal.

*The first step is to eliminate the samples with SiO2>72% and MgO<0.5% or >6.0% in each subset, thereby removing those that exceed the range of original data fitting the formula*

*The third step is to eliminate the subset whose average Rb/Sr value is greater than 0.35, in order to filter out samples that have been strongly affected by fractionation.*

*In the second step, samples of alkaline and shoshonitic basalts derived from the enriched lithospheric mantle are excluded, as these rocks typically have abnormally high Ce/Y ratios (>4), rendering the results unusable.*

8) Whether tectonics or related magmatism of extension have been reported in the region in the third stage? It is lack of evidence for post-collisional environment.

During this period, a large amount of A-type granite developed in the Longshoushan, along with mafic dikes formed in an intraplate extensional environment, indicating that the Longshoushan was in a post-collision extensional environment during the 435–410 Ma period. This viewpoint is discussed in Zeng et al., 2016 JAES, 2021 GM and 2021 Geochemistry. Our study reports evidence for crustal thinning, which is also an indication of extension.

**Other comments and suggestions:**

9) Line 55-56: "Therefore, there is still considerable debate regarding the tectonic background and tectonic evolution of the Longshoushan during the Early Paleozoic." This sentence can be deleted.

Revised

10) Line 134: "exhibited" change to "exhibit".

Revised

11) Line 381: "HREE distribution curves" change to "HREE distribution patterns".

Revised

12) Line 485: "were developed" change to "intruded".

Revised

13) Line 489: "that" change to "than"

Revised

---

## Author Comment (AC3)

**CC1**

Dear Adar Glazer:

Thank you for your comments concerning our manuscript "Petrogenesis of Early Paleozoic I-type granitoids in the Longshoushan and implications for the tectonic affinity and evolution of the southwestern Alxa Block" (EGUSPHERE-2024-1145). Those comments are all valuable and very helpful for revising and improving our paper. We have studied the comments and suggestions carefully and have made corrections. We hope our revisions meet with your approval. Below, the comments are addressed point by point. However, I recommend you read the PDF attached (Supplement), as it will be more convenient for reading and contains images.

**Major comments**

1) Some of the conclusions of this manuscript are based on geochemical analyses for which statistical measures are not provided. For example, linear correlations are sometimes referred as "good" without mentioning the R-squared values. Adding R-squared values would assure the readers that correlations are actually good and improve the reliability of this manuscript. Also, in Figure 13, KDE bandwidth and histogram bin width are not reported.

Thank you for your comment. It is problematic to use R-squared values with the number of samples involved – although for the record, they are high (Th-Rb: R-squared values=0.89; P2O5-SiO2: R-squared values=0.98; the R-squared values in Harker diagrams > 0.95; Rb/V-1/V and La/Cr-1/Cr: R-squared values=0.79–0.96). We have edited the text to refer to "trends" rather than "correlations". The word "good" is not used in the text; instead we describe trends as positive or negative. The histogram bin width value has been added to Figure 13. Figure 13 was created using 'Isoplot4.15,' and there is no setting for KDE bandwidth.

2) Based on the εHf(t)-age pattern of zircons, the authors suggest that "the Longshoushan Complex is most likely the crustal source of the granitoids in this

study". As most of the zircons fall outside the εHf(t)-age field of Longshoushan Complex rocks, more evidence should be provided to support this argument.

Thank you for your comment. We have rewritten this sentence to explain the reason:

*"In crust-mantle mixing processes, crust-derived magmas typically have lower εHf(t) values. Therefore, lower εHf(t) values can roughly reflect the composition of the crustal source. As shown in Fig. 10, some spots with lower εHf(t) values from both rock bodies fall within the evolutionary trend line of the Early Precambrian basement strata, known as the Longshoushan Complex, into which the studied plutons are intruded."*

3) In several cases methods are described outside the 'Methods' section making the manuscript a bit complicated and tiring for reading.

We have moved this part of the content: 'In this paper, the $^{206}Pb/^{238}U$ age and $^{207}Pb/^{206}U$ ages are determined for younger zircons (<1000 Ma) and older grains (>1000 Ma)' to the Methods section.

4) Please avoid the use of acronyms. That would make the article more accessible to the readers.

Considering the Reviewer's suggestion, We have changed the abbreviations in the manuscript to their full names, such as NOB, SOB, NQVA.

**Minor comments**

5)Line 39: south > southern

Revised.

6) Line 40: Add a reference to your map when introducing the study area.

Thank you for your comment. We have added a reference to the map in the introduction section.

7) Line 54-55: change ca. to ~ or use one of them throughout the whole manuscript.

Thank you for your comment. We have standardized 'ca.' to '~'.

8) Line 105: "with a small amount of gabbro also present" > with subordinate Gabbroic rocks.

Revised.

9) Line 109: "The investigated plutons in this study" > In this study we investigated two plutons located…

Revised.

10) Line 112: Duplication of "biotite quartz schist".

Revised.

11) Line 147: CJ-1 > GJ-1

Revised.

12) Line 153: "has the values of" – values of what? Please state.

We sorry for the ambiguity in this sentence; The revised content is as follows:

*During our analyses, the value of Plešovice, 91500 and GJ-1 were 0.282472–0.282495, 0.282302–0.282314 and 0.282024–0.282032 respectively, consistent with their recommended values (Plešovice: 0.282482 ±23; 91500: 0.282308 ±106; GJ-1: 0.282010 ±89, Zhang et al., 2020).*

13) Line 154: It would be much more convenient for the reader to omit all these zeros right of the decimal point, e.g. "Plešovice: 0.282482 ±23".

Thank you for your comment. The revised content is as follows:

*During our analyses, the value of Plešovice, 91500 and GJ-1 were 0.282472-0.282495, 0.282302-0.282314 and 0.282024-0.282032 respectively, consistent with their recommended values (Plešovice: 0.282482 ±23; 91500: 0.282308 ±106; GJ-1: 0.282010 ±89, Zhang et al., 2020).*

14) Line 185: slight > slightly

Revised.

15) Line 191: and in many other cases: use zircons instead of "spots".

Thank you for your comment. Using 'spot' is more accurate because some zircons are quite complex, with inherited cores and metamorphic rims. Therefore, a single zircon may have multiple spots, and each spot can only represent the zircon characteristics of its specific area, rather than the entire zircon.

16) Line 201: "which converts to εHf(t)" > with εHf(t) of

Revised.

17) Line 202: what do you mean by "using the weighted mean age"? Each zircon has its crystallization and Tdm age. please clarify.

Theoretically, the weighted mean age of the co-magmatic zircons can more accurately define the crystallization age of the co-magmatic zircons. So, we use the weighted mean age of the co-magmatic zircons to calculate the εHf(t) and Tdm age for each co-magmatic zircons.

18) Line 207: "In this paper, the $^{206}Pb/^{238}U$ age and $^{207}Pb/^{206}U$ ages are determined for younger zircons (<1000 Ma) and older grains (>1000 Ma)." – should be moved to methods.

We have moved this content to the methods section.

19) Line 208-211 (and in other cases): referring to spots #1/2/3… is very confusing and not necessary. please consider it again. You can potentially just say "Among them, three zircons are weakly luminescent…"

Revised

20) Line 216: Twelve zircon spots > Twelve zircons

Revised

21) Line 218: "The $^{176}Hf/^{177}Hf$ ratio of #6 (825 Ma) is 0.281812, which converts to

εHf(t) value of -16.07, and TDM$_2$ of 2722." > One older zircon (825 Ma) has a $^{176}$Hf/$^{177}$Hf ratio of…

Revised

22) Line 234: would be useful to add a Th/U vs. Age diagram when discussing the implications of Th/U values. Also, you state that the ages of zircons from the second group represent the timing of metamorphism. What are their ages? Do their ages correspond to any metamorphism event known in the region?

Thank you for pointing this out. We have added a Th/U vs. Age diagram. We also included the metamorphic ages in the article and discussed their significance. The content is as follows:

*The spots from the II-type zircons have ages of 1847 –1894 Ma, which is consistent with the age of the metamorphic events in the Longshoushan area during the Paleoproterozoic (Gong et al., 2016; Zeng et al., 2018)*

[Figure]

23)Line 280: "Thus, the monzogranite and K-feldspar granite are not A-type granites" and are more compatible with being I or S-type granites.

Revised

24) Line 284: Those correlations seem to be very weak. Please provide R-squared values.

Thank you for your valuable comment. The R-squared values range from 0.89 to 0.98. We have added these R-squared values in the MS to demonstrate a significant

correlation.

25) Line 293: Also here, I wouldn't say that these are "clear linear correlations" as they are not so clear. Please provide R-squared values.

We have added a description of the R-squared values in MS, all of which are greater than 0.95.

26) Line 311: "The εHf(t) values of the monzogranite" / "while those of the K-feldspar granite" – εHf(t) values are of zircons, not whole rock.

Revised

27) Line 312: "a large range of variation" > a large range of εHf(t)

Revised

28) Line 318: Please provide R-squared values.

We have added a description of the R-squared values in the MS.

29) Line 320: Add a few words on the Xijing clinopyroxene diorite and Jiling granite and how they relate to the studied plutons. That would make the manuscript more accessible to the international community.

Thank you for pointing this out. They are both located in the Longshoushan area, and we have clarified this relationship in the MS.

30) Line 328: mantle-derived and crust-derived > mantle and crust-derived

Revised

31) Line 338: "…Longshoushan Complex, into which the studied plutons are intruded.

Revised

33) Line 340: Actually, most of the zircons from your samples fall outside the

εHf(t)-age field of the Longshoushan Complex, so arguing based on εHf(t)-age data that "the Longshoushan Complex is most likely the crustal source of the granitoids in this study" is inaccurate.

Thank you for your comment. We have rewritten this sentence to explain the reason:

*"In crust-mantle mixing processes, crust-derived magmas typically have lower εHf(t) values. Therefore, lower εHf(t) values can roughly reflect the composition of the crustal source. As shown in Fig. 10, some spots with lower εHf(t) values from both rock bodies fall within the evolutionary trend line of the Early Precambrian basement strata, known as the Longshoushan Complex, into which the studied plutons are intruded."*

34) Line 356: During the Paleozoic to Mesozoic > During the Paleozoic-Mesozoic.

Revised

35) Line 359: "which includes areas ~2000 km to both the east and west of the Alxa Block" > which extends ~2000 km east and west of…

Revised

36) Line 377: is > are

Revised

37) Line 381: some scholars > various authors

Revised

38) Line 387-389: awkward phrasing. That observation has far-reaching implications for the geology and tectonic evolution of your study area. Please rephrase so it is clearer for the reader.

Thank you for your comment. The revised content is as follows:

*Figure 12 indicates that magmatism in the Alxa Block started later than 500 Ma which is later than in other parts of the Central China Orogenic Belt to the south. In*

*addition, the Longshoushan area lacks the twin peaks in Early Paleozoic ages for magmatism found in other areas such as the North Qilian Orogenic Belt and Central Qilian Block (Allen et al., 2023).*

39) Section 6.4.2: that section should be modified so it becomes smoother and clearer. Methods should be moved to the 'Methods' section or just be cited and the results of your crustal thickness analysis should become more prominent.

This is the data processing workflow, not the testing method. Additionally, this section processes not only the data obtained from this study but also a majority of previous data. Therefore, we did not move this section to the Methods.

40) Line 425: which statistics?

Thank you for your comment. Figure 2 provides a statistical of the rock ages. The revised content is as follows:

*A notable feature of the Longshoushan is the large volume of Late Ordovician-Silurian magmatic rocks (Fig. 2). Based on the statistics and analysis of U-Pb ages and geochemical data of these rocks.*

41) Line 426: can be divided > can be defined.

Revised

42) Line 441: tectonic background > tectonic setting

Revised

43) Line 457: calc-alkaline granitic magmatism

Revised

**Figures**

44) Figure 1: would be useful to add an inset of the world map for orientation. Please highlight your study area.

For the world, the Alxa Block is too small, so we did not include a world map; instead, we chose a map of China.

In Figure 1a, we represent the Alxa Block in bold red font. In Figure 1b, we added a star to indicate the study area.

[Figure]

45) Figure 2: add in the figure caption some reference to the geochronology you present. Are these zircon U-Pb ages? Whole rock Rb-Sr? Please describe. Highlight your study area in Figure 2a. 'Mafic rocks' appear twice in the legend.

Thank you for your comment. We have revised the title of Figure 2 to indicate that the data come from zircon U-Pb dating. In Figure 2a, we added a bold blue box to represent the study area. One legend of Mafic rocks has been deleted.

[Figure]

Figure 2 (a) Simplified geological map of the southwestern Alxa Block (Wang et al., 2020); (b) Simplified geological map of the east of the Longshoushan area. Data are in Supplementary materials Table S1 (All age data were obtained using the zircon U-Pb method).

46) Figure 3: mineral abbreviations and scale bars should be highlighted.

Thank you for your comment. We added a white background to the abbreviations and scale bars.

[Figure]

47) Figure 11: what is the meaning of the red arrow, please describe.

We explained the meaning of the red arrows in the figure caption.

*Figure 10 Zircon εHf(t)-age (Ma) diagram for samples in this study and published data for the region. The source of the published data can be found in Supplementary materials Table S1 (The red arrow represents the variation trend of εHf(t)).*

48) Figure 13: please provide KDE bandwidth and histogram bin width.

Thank you for your comment. The histogram bin width value has been added to Figure 13. Figure 13 was created using 'Isoplot4.15,' and there is no setting for KDE bandwidth.

The legend for open circles has been added, as follows:

[Figure]

Calculated from median Sr/Y
Calculated from highest Ce/Y
magmatic rock samples

50) Figure 15: would be more informative to give a different color to ophiolite belts. Add in legend the black arrows, and highlight the pink cross section paths on map view.

We have bolded the pink cross-section paths. In the legend, we explained the meaning of the black arrows. The ophiolite is included in the subduction-accretion complex, with the South Ophiolite Belt and Northern Ophiolite Belt being the names of these two tectonic units, as cited from Allen et al. (2023 ESR).